palaeontology/ecology

Ankylosauria, diet, Cretaceous, cololite, Clearwater Formation, Canada

**Author for correspondence:**
Caleb M. Brown
e-mail: caleb.brown@gov.ab.ca

# Dietary palaeoecology of an Early Cretaceous armoured dinosaur (Ornithischia; Nodosauridae) based on floral analysis of stomach contents

Caleb M. Brown[1], David R. Greenwood[2],
Jessica E. Kalyniuk[2], Dennis R. Braman[1],
Donald M. Henderson[1], Cathy L. Greenwood[2]
and James F. Basinger[3]

[1]Royal Tyrrell Museum of Palaeontology, Drumheller, Alberta, Canada T0J 0Y0
[2]Department of Biology, Brandon University, Brandon, Manitoba, Canada R7A 6A9
[3]Department of Geological Sciences, University of Saskatchewan, Saskatoon, Saskatchewan, Canada S7N 5E2

 CMB, 0000-0001-6463-8677; DRG, 0000-0002-8569-9695

The exceptionally well-preserved holotype of the armoured dinosaur *Borealopelta markmitchelli* (Ornithischia; Nodosauridae) from the Early Cretaceous (Clearwater Formation) of northern Alberta preserves a distinct mass within the abdominal cavity. Fourteen independent criteria (including: co-allochthony, anatomical position, gastroliths) support the interpretation of this mass as ingested stomach contents—a cololite. Palynomorphs in the cololite are a subset of the more diverse external sample. Analysis of the cololite documents well-preserved plant material dominated by leaf tissue (88%), including intact sporangia, leaf cross-sections and cuticle, but also including stems, wood and charcoal. The leaf fraction is dominated (85%) by leptosporangiate ferns (subclass Polypodiidae), with low cycad–cycadophyte (3%) and trace conifer foliage. These data represent the most well-supported and detailed direct evidence of diet in an herbivorous dinosaur. Details of the dietary palaeoecology of this nodosaur are revealed, including: selective feeding on ferns; preferential ingestion of leptosporangiate ferns to the exclusion of Osmundaceae and eusporangiate ferns such as Marattiaceae; and incidental consumption of cycad–cycadophyte and conifer

leaves. The presence of significant (6%) charcoal may represent the dietary use of recently burned conifer forest undergoing fern succession, early evidence of a fire succession ecology, as is associated with many modern large herbivores.

# 1. Introduction

Dinosaurs dominated terrestrial landscapes for over 150 Myr, and included a diversity of herbivorous forms, from ornithischians to sauropods and many theropods [1,2]. Despite the importance of herbivorous dinosaurs in Mesozoic ecosystems, there is very little direct evidence of diet. What large herbivorous dinosaurs ate has implications for our understanding of how Mesozoic terrestrial ecosystems functioned, and the physiology and ecology of these animals.

Across modern ecosystems, large terrestrial herbivores, and specifically megaherbivores (*sensu* [3]; i.e. herbivores with a mass greater than 1000 kg), have disproportionate effects on the landscapes they occupy and are termed 'keystone herbivores' [3–7]. In modern ecosystems, this ecological guild is occupied exclusively by mammals [3]. During the Mesozoic, the megaherbivore niche was occupied by dinosaurs, where multi-ton herbivores evolved independently a minimum of five times within Dinosauria (Sauropodomorpha: Triassic; Thyreophora: Triassic/Jurassic; Iguanodontia: Jurassic; Therizinosauria: Cretaceous; and Ceratopsidae: Cretaceous) [8].

Dinosaur megaherbivores greatly exceeded mammal megaherbivores in both temporal duration (150 Ma versus 40 Ma) and body mass [8,9], and a disproportionate 'keystone' effect of megaherbivores may be expected for Mesozoic dinosaurs. To test this, sustained research in dinosaur megaherbivore palaeoecology has attempted to analyse the diet and ecological interactions of these animals [2,10–12], with multiple authors hypothesizing, and testing, dinosaur megaherbivore and plant coevolutionary patterns [13–16]. Despite this broad interest, direct data on the diets of herbivorous dinosaurs are very limited, and researchers have largely inferred diet based on factors including: flora availability and energy/nutrient content [12,17–19]; dinosaur–plant associations [20–22]; jaw biomechanics [1,23–27]; tooth wear [26,28–30]; posture and feeding height [31,32]; and stable isotopes [33,34].

Drawing direct parallels between modern mammal and Mesozoic dinosaur megaherbivores is hindered due to major differences in both the animals and available plant food sources. Herbivorous dinosaurs possess drastically different dental and masticatory anatomy compared to mammals, and largely unknown thermal and digestive physiology. Although some dinosaur megaherbivores evolved complex dentitions capable of mastication [23,35], and potentially on par with mammals [36,37], many dinosaur megaherbivore clades possessed simple teeth capable of cropping plants, but with inefficient capacity for mastication [38]. Broad dietary hypotheses have been suggested for many groups, but these have proven difficult to test. Furthermore, the foliage available as diet for these dinosaur megaherbivores, largely ferns and gymnosperms (e.g. conifers, cycads–cycadophytes), was quite different from that of modern megaherbivores, for which angiosperms, and in particular grasses, make up a dominant portion of large herbivore diets [12,18,22,39,40].

Direct evidence of diet in herbivorous dinosaurs is rare and comes in the form of coprolites (fossil faeces), and even rarer still, cololites (fossil stomach or intestinal contents). Coprolites of herbivorous dinosaurs [41–45] often provide little dietary information and are difficult to match to the trace maker [46]. Although cololites (or other fossil gastrointestinal contents) have been reported in many herbivorous taxa, nearly all of these reports have not held up under closer analysis [12,47–49] (table 1), and in many cases do not provide data beyond indeterminate plant fragments [47]. Within herbivorous dinosaurs, putative cololites or other preserved stomach contents have been reported for three major groups: Sauropoda, Ornithopoda (largely Hadrosauridae) and Thyreophora (largely Ankylosauria) (table 1).

Several reports of putative cololites from Sauropoda have been published, [50–52]; but all of these are now regarded as unlikely to be stomach contents [12,47] (table 1).

Multiple cases of fossil gastrointestinal contents in Hadrosauridae have been reported [47,49,53,54]; although most of these are now viewed as unlikely to be truly gastric/intestinal in origin [12,48,49], or are viewed as equivocal at best [47,49]. The most recent and detailed account is that of a mummified specimen of *Brachylophosaurus canadensis* [47]. However, even if verified, this provides little in the way of novel dietary and digestive information other than that it may indicate the presence of chewed leaves (table 1).

To date, armoured dinosaurs have preserved a more credible account of stomach contents than any other group of herbivorous dinosaurs (table 1). The small ankylosaur *Kunbarrasaurus ieversi* from the

**Table 1.** List of reported stomach contents for herbivorous non-avian dinosaurs. Specimens for which criteria supporting attribution of material to stomach contents in electronic supplementary material, table S1 are indicated by '*'.

| major clade | species | specimen | provenance | contents | confidence |
|---|---|---|---|---|---|
| Saurischia: Sauropoda | unknown | unknown | Morrison Formation (Kimmeridgian), Utah, USA | twigs, branches, bone fragments and a tooth [50] | not supported [12,47] |
| Saurischia: Sauropoda | unknown | unknown | Morrison Formation (Upper Jurassic, Kimmeridgian-Tithonian), Wyoming, USA | carbonaceous material, mature stems, bits of leaves, other plant matter [51] | not supported [12] |
| Saurischia: Sauropoda | unknown | unknown | Morrison Formation (Upper Jurassic, Kimmeridgian-Tithonian), Wyoming, USA | 'Vergeben wurden'—plant fossils [52] | not supported [12] |
| Ornithischia: Hadrosauridae | Edmontosaurus annectens | AMNH 5060 | Lance Formation (late Maastrichtian), Wyoming, USA | undescribed [53] | uncertain |
| Ornithischia: Hadrosauridae | Edmontosaurus annectens | NMS 4036 | Lance Formation (late Maastrichtian), Wyoming, USA | Cunninghamia needles, conifer and deciduous (angiosperm) branches, and small seeds or fruits [54] | not supported [12,47,48] |
| Ornithischia: Hadrosauridae* | Corythosaurus casuarius | TMP 1980.040.0001 | Dinosaur Park Formation (late Campanian), Alberta, Canada | twigs and stems of gymnosperms and angiosperms, bark, seeds and charcoal [49] | equivocal [47] |
| Ornithischia: Hadrosauridae* | Brachylophosaurus canadensis | JRF 115H | Judith River Formation (Campanian), Montana, USA | mm-scale indeterminate leaf fragments [47] | equivocal |
| Ornithischia: Ankylosauria* | Kunbarrasaurus ieversi | QM F18101 | Toolebuc Formation (Albian) Queensland, Australia | irregular strands of vascular tissue, spheroidal seed-bearing organs, possible sporangia [55,56] | well supported [47] |
| Ornithischia: Thyreophora?* | Isaberrysaura mollensis | MOZ-Pv 6459 | Los Molles Formation (Toarcian-Bajocian), Neuquén Province, Argentina | seeds, including Cycadales (Zamineae) and indeterminate smaller, platyspermic seeds [57] | supported |
| Ornithischia: Nodosauridae* | Borealopelta markmitchelli | TMP 2011.033.0001 | Wabiska Member, Clearwater Formation (Albian), Alberta, Canada | diversity of well-preserved floral taxa and tissues: see results | well supported |

Early Cretaceous of Australia is described as preserving a cololite within the abdominal cavity [55,56]. Several criteria support the conclusion that this material represents a cololite. Although well supported, much of the preserved material consists of non-diagnostic fragments of vascular tissue, indeterminate seed-bearing organs, and possible sporangia, which does little to inform on diet. The recently described ornithischian *Isaberrysaura mollensis* from the Early Jurassic of Argentina (originally described as a basal neornithischian [57], but see recent re-evaluations suggesting it is a Thyreophoran [58,59]), also preserves putative stomach contents. These are described as Cycadales (Zamiineae) seeds as well as smaller indeterminate seeds. However, the description of the stomach contents is limited, and taxonomic stability of the dinosaur uncertain, and more refined assessment of the stomach contents is needed. Finally, Ji *et al*. [60] report the unexpected finding of ingested fish in a small-bodied early Cretaceous ankylosaur from China, suggesting a potential piscivorous, or at least omnivorous, diet.

Here, we document, and quantify, an unequivocal and exceptionally well-preserved cololite from the Early Cretaceous dinosaur *Borealopelta markmitchelli* (Ornithischia, Nodosauridae) [61]. The analysis provides the most detailed account of direct evidence of diet in an herbivorous Mesozoic dinosaur, and informs the palaeoecology of armoured dinosaurs.

# 2. Material and methods

## 2.1. Specimen and geological context

The holotype specimen of the recently described armoured dinosaur *Borealopelta markmitchelli*, TMP 2011.033.0001 is exceptional in preserving soft tissue, including scales and keratinous coverings of the bony osteoderm armour across the body, while also maintaining its three-dimensional shape [61]. In addition to the animal tissue, a large spheroid mass within the abdominal cavity is also preserved (figures 1 and 2). This mass is composed of a distinct matrix, dominated by organic and inorganic inclusions, and is interpreted as a cololite.

*Borealopelta markmitchelli* was recovered near the base of a 3 m thick greenish-grey, very fine- to medium-grained glauconitic sandstone unit within the marine Wabiskaw Member of the fully marine Clearwater Formation, approximately 8 m above the underlying McMurray Formation. This same restricted glauconitic sandstone unit has yielded abundant articulated marine reptiles, including the ichthyosaur *Athabascasaurus bituminous* [62], the plesiosaur *Nichollssaura borealis* [63], a polycotylid [64] and the elasmosaur *Wapuskanectes betsynichollsae* [65]. This fossil-rich unit was deposited in the lower shoreface to offshore transition zone, between the fair- and storm-weather wave base [66]. Combined data from ammonite, palynological and foraminiferal biostratigraphy indicate an Early Albian age [62]. Geographically, the specimen was recovered from the Suncor Millennium Mine (open pit, oil sands) in northern Alberta (UTM; 12 U; 478 446 m E; 6,315,224 m N, WGS84).

## 2.2. Criteria for assessing support for stomach contents

In order to develop a more objective, or at least more explicit, rationale for evaluating the evidence supporting putative cololites (preserved stomach contents), a series of independent criteria were established (table 2). Criteria applied to evaluate or validate putative stomach contents build on those suggested by Molnar & Clifford [55], as well as some criteria, either explicit or implicit, of Currie *et al*. [49], Mayr [69], Tweet *et al*. [47], O'Keefe *et al*. [67] and Druckenmiller *et al*. [68]. In total, 16 criteria are used to evaluate published accounts of cololites in herbivorous dinosaurs (table 2). Most criteria are developed to be generally applied, but some criteria (e.g. co-occurrence with gastroliths) are not applicable to all taxa. Similarly, these criteria are developed for herbivorous taxa, and while several criteria are independent of diet (e.g. co-allochthony, anatomical position, exceptional preservation), others are diet-dependent (e.g. co-occurrence with gastroliths, mastication), and may not be applicable across all diets and taxa. Similarly, additional criteria may be relevant for carnivores (e.g. tooth-marked bone, acid etching of bone), but are not applicable for herbivores.

## 2.3. Palynology

Palynological preparations were made of samples of the cololite and of the external matrix in order to compare the composition of the last meal preserved within the cololite with the regional and time-averaged vegetation recorded in the surrounding sediment. The samples were processed using the

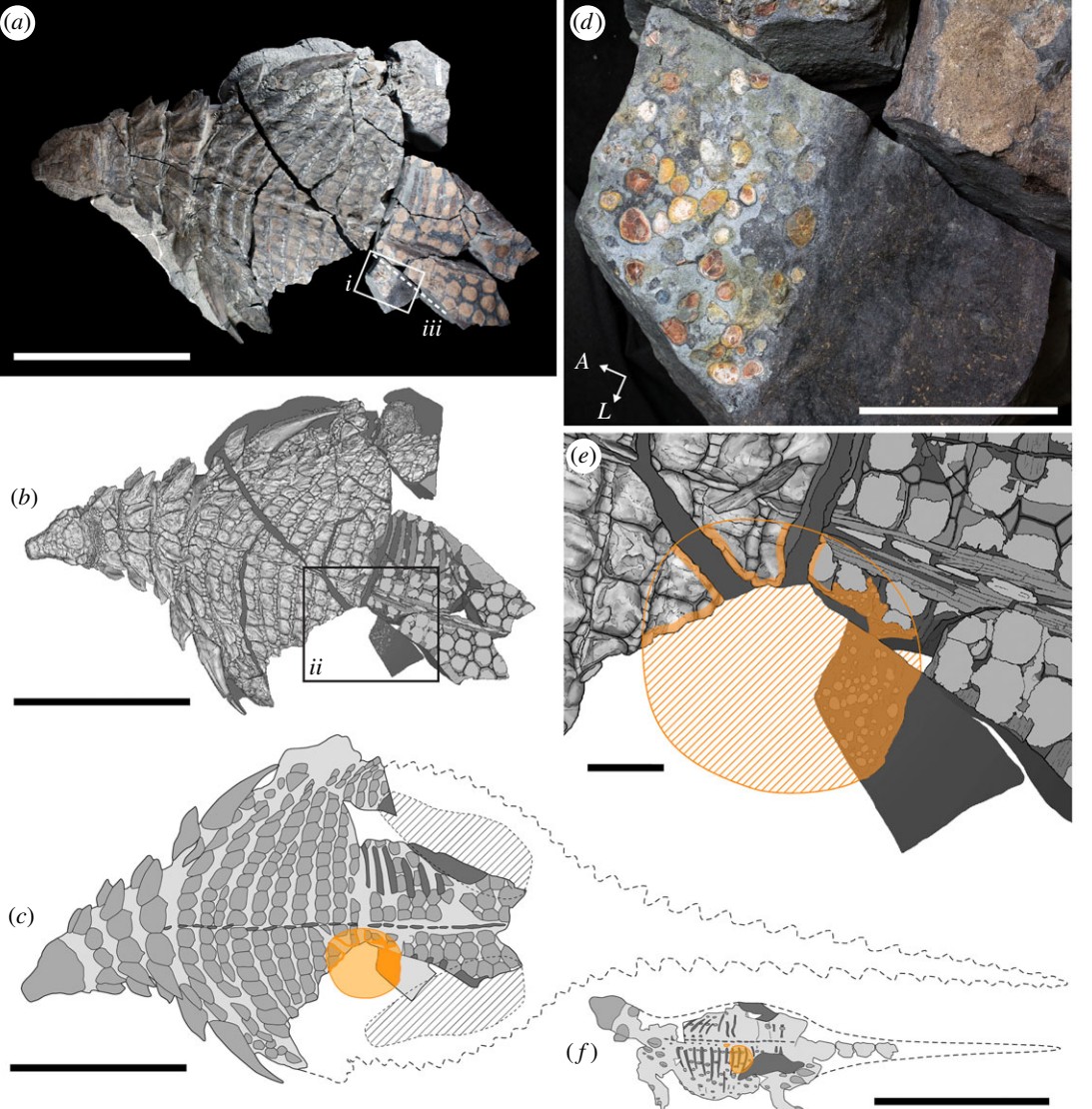

**Figure 1.** Location of abdominal mass, including stomach contents (cololite), within the well-preserved nodosaur *Borealopelta markmitchelli* (TMP 2011.033.0001). Photograph (*a*) and scientific line drawing (*b*) of the specimen in dorsal view. Schematic drawing (*c*) of specimen showing position and extent of abdominal mass, as well as extrapolated body outline. Inset (*d*) of *i*, showing close up photograph of dorsal view of posterior margin of abdominal mass. Inset (*e*) of *ii*, showing detailed map of extent of abdominal mass. (*f*) Schematic drawing of *Kunbarrasaurus ieversi* (GM F18101) scaled to (*c*), showing relative size and positon of cololite. Solid orange, observed cololite; hatched orange, inferred cololite. *A*, anterior; *L*, lateral. Scale bars in (*a,b,c,f*) are 1 m, and in (*d,e*) are 10 cm.

standard palynological techniques [70], with the resulting slides assigned the palynological series number TMP 2017.205.

## 2.4. Microscopic palaeobotanical analysis of cololite

### 2.4.1. Histological sections

Seven thin sections of the cololite were prepared in order to microscopically analyse its contents, including mineralogy of the matrix and gastroliths, as well as to allow analysis of any preserved organic matter (electronic supplementary material, figure S1*a–g*). Additionally, two thin sections sampling transects from the external matrix into the cololite were also prepared, to allow for investigation of this transition (electronic supplementary material, figure S1*h–j*). Thin sections were prepared using standard techniques: embedded in clear resin under vacuum; cut to size; mounted on

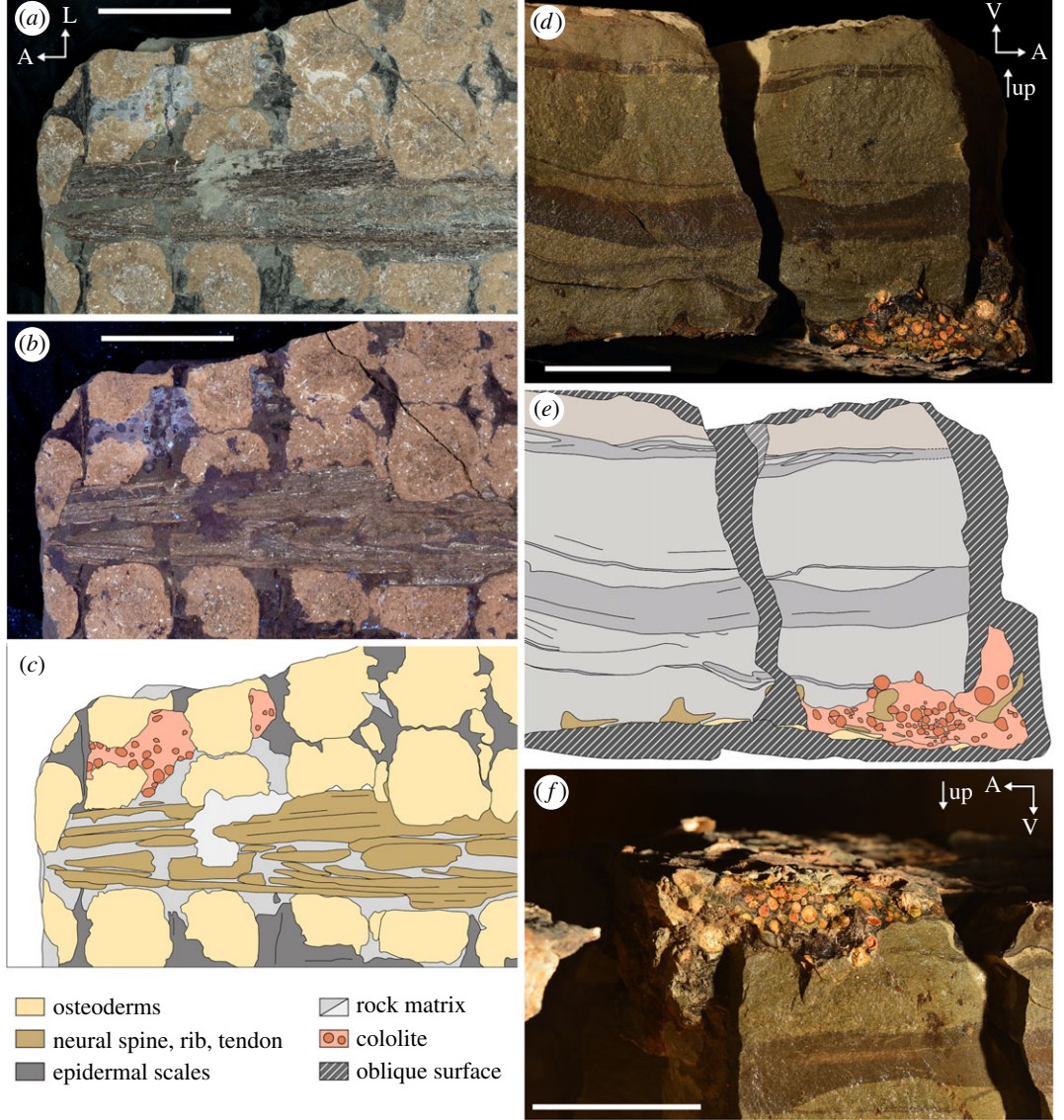

**Figure 2.** Images of preserved abdominal mass of TMP 2011.033.0001. Visible (*a*) and UV fluorescence (*b*) photographs and interpretive drawing (*c*) of block F, in the plain of the sacral armour (i.e. frontal plain). Wetted cross-polarized light photograph (*d*) and interpretive drawing (*e*) of the abdominal mass in lateral view (in stratigraphic position). (*f*) Wetted cross-polarized light photograph of abdominal mass in dorsolateral view (in anatomic position). Orientations: A, anterior; L, lateral; V, ventral. All scale bars equal 10 cm. Legend applies to both (*c*) and (*e*).

glass slides; and thinned until desired translucence was achieved. As a result of being based on *ex situ* hand samples, the thin sections do not have known anatomical or sedimentological orientations.

### 2.4.2. Palaeobotanical analysis of material

Most of the plant material was present as small fragments typically sheared at oblique angles, obscuring tissue character and often lacking taxonomically diagnostic characters, necessitating a morphological category approach. Quantitative microscopic analysis of five of the slides used an Olympus BX51 compound microscope with Z-stack computerized imagery and 26 categories based on gross morphological criteria (table 3). For the analysis and interpretation, these categories were subsequently aggregated to yield more ecologically meaningful inclusive groupings.

### 2.4.3. Taxonomy

Taxonomic identifications of leaf material, where possible, were made using diagnostic epidermal/cuticular anatomy. Clubmoss (class Lycopodiopsida: Lycopodiaceae and Selaginellaceae) and fern (class

**Table 2.** List of criteria to support the attribution of material to dietary stomach contents in fossil material.

| no. | criteria | details |
|---|---|---|
| 1 | co-allochthonous | co-occurrence of food items and animal in non-habitat host rock (modified from Molnar & Clifford [55]) |
| 2 | anatomical position A | food items enclosed within three-dimensional body cavity (modified from Molnar & Clifford [55]) |
| 3 | anatomical position B | food items in appropriate position for stomach/intestines (modified from Molnar & Clifford [55]) |
| 4 | exceptional preservation | other tissues (e.g. skin) well-preserved (modified from Tweet et al. [47]) |
| 5 | size uniformity | uniformity of food item size (modified from Molnar & Clifford [55]) |
| 6 | mastication of material | cleanly sheared margins (modified from Molnar & Clifford [55]) |
| 7 | gastrolith association | food items associated with gastroliths (modified from O'Keefe et al. [67]) |
| 8 | mass mineralogy/ sedimentology | mass distinct from surrounding matrix (modified from Tweet et al. [47]) |
| 9 | mass margin | defined margin, organic envelope (modified from Druckenmiller et al. [68]) |
| 10 | mass shape | three-dimensional spheroid or oblong mass (new) |
| 11 | content restricted | food items localized internally, and absent in external matrix (new) |
| 12 | concentration | unusual concentration, rarity of food items (modified from Mayr [69]) |
| 13 | distinct palynomorphs | mass and external palynomorphs are distinct (modified from Tweet et al. [47]) |
| 14 | acid etching on bone | bone surface shows etching from stomach acid (modified from Currie et al. [49]) |
| 15 | geochemical | evidence of stomach enzymes, etc. (new) |
| 16 | dietary appropriate | dietary items are appropriate given independent data (modified from Currie et al. [49]) |

Polypodiopsida) sporangia were identified from *in situ* spores. Leptosporangiate ferns (subclass Polypodiidae) were identified through the presence of sporangia with a well-developed annulus: a thickened band forming along the outer margin of the sporangium. Non-leptosporangiate ferns (e.g. *Angiopteris*, Marattiaceae) and basal leptosporangiate ferns such as *Osmunda* (Osmundaceae) lack this annulus on their sporangia [71–73].

## 2.5. The contemporaneous Gates Formation (Grand Cache Member) macroflora

*Borealopelta markmitchelli* was deposited in marine sediments (i.e. Wabiskaw Member of the Clearwater Formation), and so there is no associated macroflora representing the local flora this herbivorous dinosaur fed within. The Albian coastline lay 200–400 km to the west, and so the Lower Albian Grand Cache Member of the Gates Formation [74] provides a picture of the local vegetation available to *Borealopelta* and other herbivorous dinosaurs (figure 3). We review the Grand Cache Member macroflora from an unpublished taxonomic analysis [74] and a collection in the Royal Tyrrell Museum of Palaeontology (RTMP) of fossil leaves collected from the Grand Cache Coal Mine (collected in 2015), Smokey River Coal Mine (collected in 1990) and near the McIntyre Mine (collected in 1981; detailed site information available from the RTMP database). Representative examples of key fern and gymnosperm taxa were photographed using a Canon EOS 6D digital SLR camera with 24–105 mm [1:4] lens from both the RTMP collection as well as the collection made by Wan [74] housed at the University of Saskatchewan (figure 3). Identifications follow Wan [74], except where a new unpublished taxon was proposed by Wan, in which case we have used the currently accepted generic name or species binomial.

## 2.6. Institutional abbreviations

AMNH, American Museum of Natural History, New York, USA; JRF, Judith River Foundation, Malta, Montana, USA; NMS, Naturmuseum Senckenberg, Frankfurt, Germany; TMP, Royal Tyrrell Museum of Palaeontology, Drumheller, Alberta, Canada; QM, Queensland Museum, Brisbane, Queensland,

**Table 3.** Plant fragment categories used in the analysis.

| | category name | description |
|---|---|---|
| 1 | cuticle with stomata Type A | fern or angiosperm type stomata (bean-shaped guard cells) present |
| 2 | cuticle with stomata Type B | gymnosperm type stomata (polygonal guard cells) with a prominent polar flange |
| 3 | cuticle type 1 | epidermal cells with low-amplitude and high-frequency sinuous lateral walls |
| 4 | cuticle type 2 | epidermal cells with high-amplitude and low-frequency sinuous lateral walls |
| 5 | cuticle type 3 | epidermal cells with high-amplitude and high-frequency sinuous lateral walls |
| 6 | leaf epidermis with cellular material | typically has stomata present; epidermal cells and often also underlying parenchyma (palisade and/or mesophyll) present |
| 7 | leaf cross-section with thickened cells | recognizably a leaf fragment showing one or more of parenchyma, vascular cells or epidermal cells in addition to sclerenchyma |
| 8 | leaf cross-section without thickened cells | recognizably a leaf fragment showing one or more of parenchyma, vascular cells, or epidermal cells with no presence of sclerenchyma |
| 9 | leaf cross-section undifferentiated | recognizably a leaf fragment showing one or more of parenchyma, vascular cells or epidermal cells but where further differentiation is not possible |
| 10 | leaf cross-section undifferentiated 'narrow' | recognizably a narrow in cross-section leaf fragment showing one or more of parenchyma, vascular cells or epidermal cells but where further differentiation is not possible |
| 11 | undifferentiated plant material (probably all leaf mesophyll) | recognizably plant tissue, typically mostly parenchyma, but not assignable to any of the other categories |
| 12 | clump of thickened cells (sclerenchyma) | sclerenchyma that is not associated with any other cells or recognizable tissue |
| 13 | wood/woody stems | plant tissue composed mostly of xylem tissue; includes woody stem cross-sections showing growth rings |
| 14 | isolated tracheids/xylem fragments | isolated individual tracheids or scattered clumps of tracheids or similar vascular tissue |
| 15 | round stem cross-section | a round piece of plant tissue that shows a discrete dermis on the outside, and internal to the dermis evidence of structure such as vascular bundles, parenchyma or other recognizably organized tissues; these may be leaf petioles or reproductive structures such as strobili |
| 16 | square stem cross-section | a square or polygonal piece of plant tissue that shows a discrete dermis on the outside, and internal to the dermis evidence of structure such as vascular bundles, parenchyma or other recognizably organized tissues; these are probably petioles or leaf mid-veins |
| 17 | stem longitudinal section | an irregularly polygonal piece of plant tissue that shows evidence of longitudinally organized tissues such as vascular bundles, parenchyma or other recognizable tissues; may include longitudinally sectioned stobili |
| 18 | stem cross-section uncertain | as for the above, but where the angle of cut prevents recognition of the shape being round or polygonal |

(*Continued.*)

**Table 3.** (Continued.)

| | category name | description |
|---|---|---|
| 19 | sporangia undifferentiated | any sporangia that cannot be placed in any of the sporangia categories (A–F) |
| 20 | sporangia type A | leptosporangiate: has a prominent annulus with dark thickened 'ridges', contains spores that are trilete and psilate with concave sides (indet.) |
| 21 | sporangia type B | leptosporangiate: contains spores that are trilete and psilate with convex sides (indet.) |
| 22 | sporangia type C | Lycopodiopsida: multiseriate band of thickened cells +/− present, contains trilete spores that are echinate and rounded (*Echinatisporis* sp.) |
| 23 | sporangia type D | leptosporangiate: contains trilete spores that are very sharply angular (indet.) |
| 24 | sporangia type E | leptosporangiate: well-developed annulus, trilete spores that are almost circular (*Deltoidospora* sp. or *Biretrisporites* sp.) |
| 25 | sporangia type F | leptosporangiate: contains trilete spores that are striate (*Cicatricocsisporites* sp.) |
| 26 | charcoal/blackened plant material | material that is black and non-shiny that otherwise is recognizably plant cells or tissue; interpreted as charcoal or fusinite |

Australia; MOZ-Pv, Museo Provincial de Ciencias Naturales 'Prof. Dr Juan A. Olsacher', Paleovertebrates collection, Zapala, Neuquén, Argentina.

# 3. Results

## 3.1. Macroscopic analysis of the cololite

The extent of the cololite can be observed in dorsal view on block K, the anteromedial extremes of block F (and counterpart), and in the broken cross-sections along the anteromedial extent of block F, posteromedial extent of block D and posterior extent of block E (figure 1) [75]. The cololite is located near the thoracosacral transition (T9–T12 osteoderm rows [75]), positioned to the left of the midline (figure 1). It is dorsally positioned within the abdominal cavity, being appressed to the ribs of rows T11–12 (figure 2*d–f*). Its shape is a vertically compressed sphere, with horizontal diameter of approximately 36 cm and a maximum height of approximately 18 cm.

The cololite is composed of a single large three-dimensional cluster of distinct spheroids (interpreted here as gastroliths) ranging from clast-supported to matrix-supported, within a light grey fine- to medium-grained sandstone matrix (figures 1*d* and 2). The spheroids are highly rounded, ranging in form from oblong to spherical, and diameter from 1.9 to 22.1 mm, and vary in colour from yellow, orange, red to brown, occasionally bearing signs of concentric rings, or surrounded by a halo of darkened matrix. The spheroids show some degree of size sorting, with those ventrally positioned (stratigraphically up) being larger, and those positioned further dorsally being smaller and more numerous (figure 2*d–f*). The matrix surrounding the spheroids is light grey with abundant, small (generally less than 2 mm), dark, organic fragments (figures 1*d* and 2). Macroscopically, and even under stereo microscopy, few details can be discerned from these organic fragments. One exception is the occurrence of a centimetre-scale woody stem observable in two cross-sections and partial longitudinal section along the margin of a hand sample block. The stem measures 34.9 mm in preserved length (truncated at both ends), with diameter of 4.5–5.8 mm.

The sediment external to the cololite is markedly different, consisting of a nearly uniformly dark grey, fine siltstone with green mottling—indicative of a prominent glauconitic component (figure 2*d–f*; electronic supplementary material, figure S3*h–j*). In all cases, the contact between the cololite and the external matrix is distinct, showing: (A) a marked transition from large spheroids to uniform and fine-grained matrix; (B) a marked transition from dense organic fragments to a matrix lacking inclusions; (C) an abrupt transition in matrix lithology (light grey fine to medium sandstone of the cololite to dark green-grey siltstone); and (D) in many cases a thin (less than 1 mm), dark, undulating and

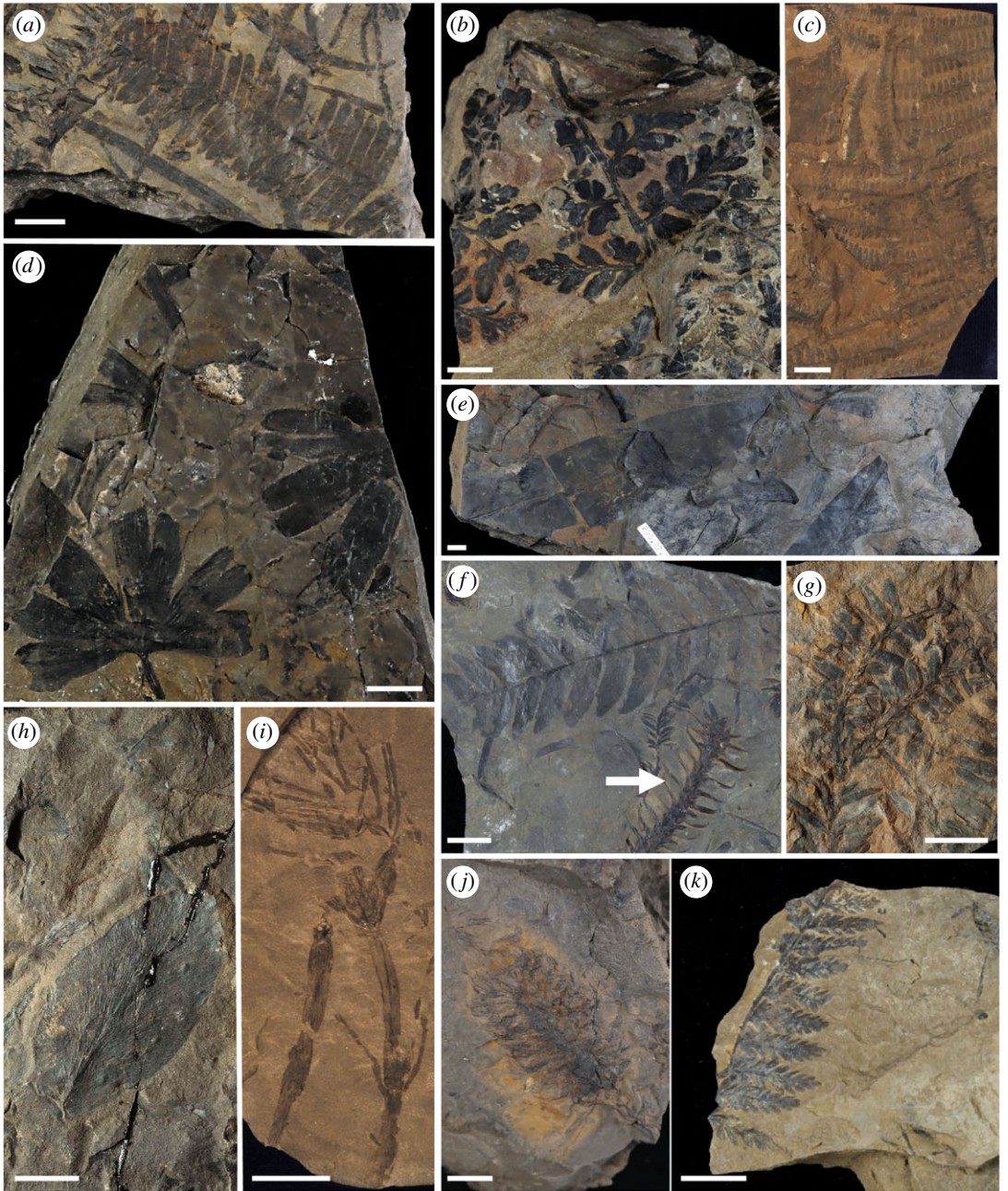

**Figure 3.** Gates Formation (Grand Cache Member) plant fossils from central Alberta. (*a*) *Pterophyllum* sp. (TMP 1990.027.0021), (*b*) *Sphenopteris* sp. (TMP 1981.055.0103), (*c*) *Gleichenites* sp. (USask 925-7273), (*d*) *Ginkgoites* sp. (TMP 1990.027.0020), (*e*) *Taeniopteris* sp. (TMP 1981.055.0006), (*f*) *Cladophlebis* sp. (top left) and *Elatides* sp. (arrow) (TMP 1981.055.0012), (*g*) *Elatides curvifolia* (TMP 2015.006.0469), (*h*) *Sagenopteris* sp. (TMP 1981.055.0033), (*i*) *Equisetites* sp. (USask 750-7557), (*j*) conifer cone (TMP 1981.055.0044) and (*k*) *Coniopteris* sp. (TMP 1981.055.0058). Scale bars = 1 cm.

discontinuous surface that may represent stomach wall or other visceral tissue (figure 2*d*; electronic supplementary material, figure S3*h–j*). The macroscopic distinction between the matrix internal and external to the cololite is further highlighted by UV light, where the internal matrix shows a higher degree of UV fluorescence (figure 2*b*) than the surrounding matrix.

## 3.2. Palynology

A total of 50 palynomorphs were recovered from the cololite and external matrix samples, including six bryophytes (i.e. moss or liverwort), 28 pteridophytes (i.e. clubmosses and ferns), 13 gymnosperms (principally conifers, and one cycad) and two angiosperms (electronic supplementary material, figure S2,

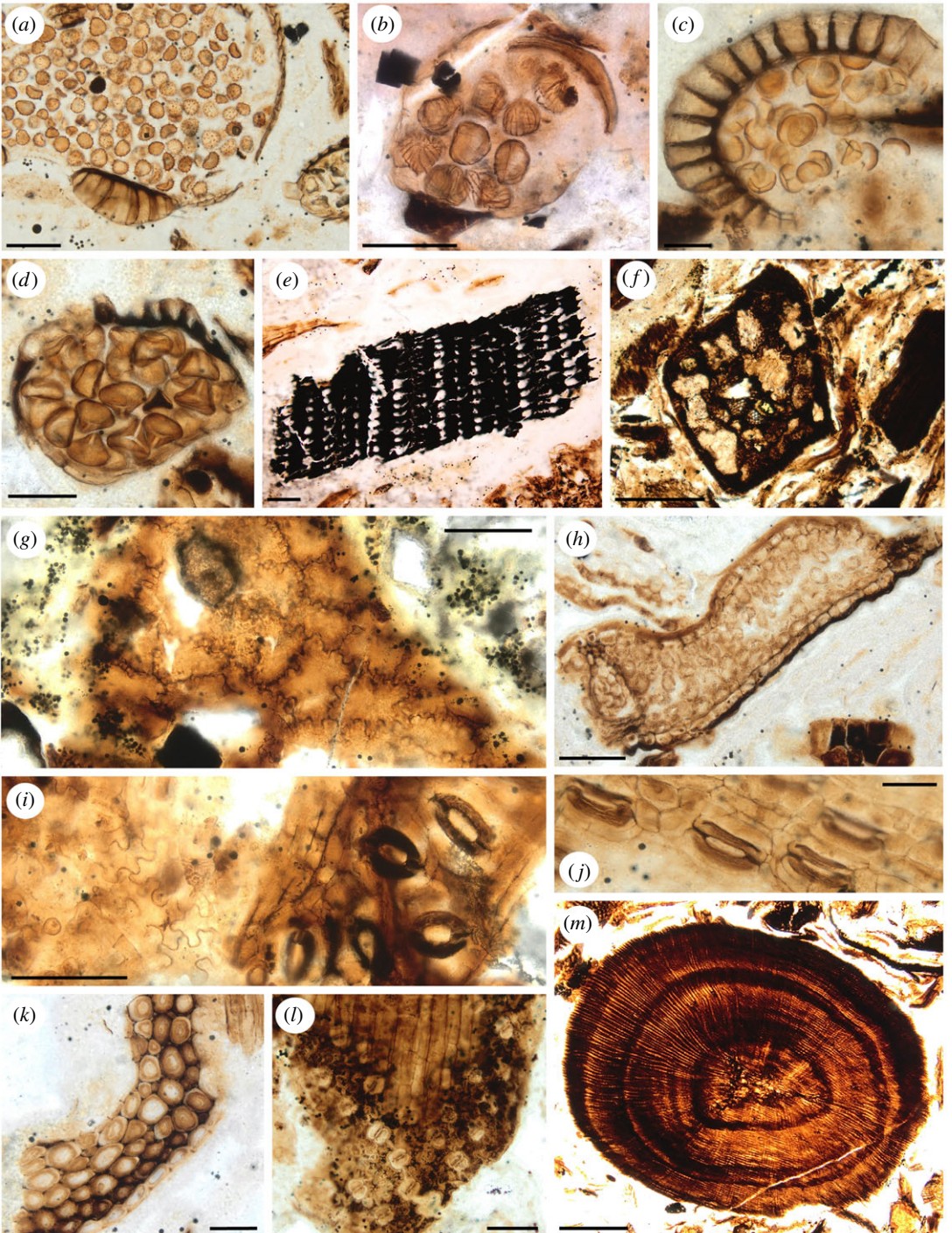

**Figure 4.** Palaeobotanical elements observed on the cololite histological slides. (*a*) Clubmoss (Lycopodiopsida) sporangium type C with *Echinatisporis* sp. (Lycopodiaceae or Selaginellaceae), (*b–d*) isolated leptosporangiate fern sporangia with spores *in situ*, (*b*) sporangium type F with *Cicatricosisporites* sp. (Schizaeaceae), (*c*) sporangium type E with *Deltoidospora* sp. (fam. indet.) or *Biretrisporites* sp. (Matoniaceae-Cyatheaceae-Dicksoniaceae), (*d*) sporangium type A (spore indet.), (*e*) charcoal/blackened plant fragment, (*f*) square stem cross-section, (*g*) cuticle without stomata displaying sinuous lateral cell walls (Type 1), (*h*) leaf cross section, (*i*) cuticle with stomata and sinuous lateral cell walls (Type 2), (*j*) cuticle with stomata Type B, (*k*) thickened cells/ sclerenchyma, (*l*) cuticle with stomata (Type A), (*m*) twig cross-section showing annual rings. (*c,d,j,k*) scale bars = 40 μm; (*a,b, e,g,h,i,l*) scale bars = 100 μm; (*m,f*) scale bars = 400 μm.

appendix 1). Pteridophyte spores were primarily leptosporangiate (Polypodiidae: 21 taxa) such as Gleicheniaceae and Schizaeaceae, as well as four species of Osmundaceae. Spores *in situ* within sporangia were also identified in the cololite sample (figures 4 and 5). The external matrix has a much

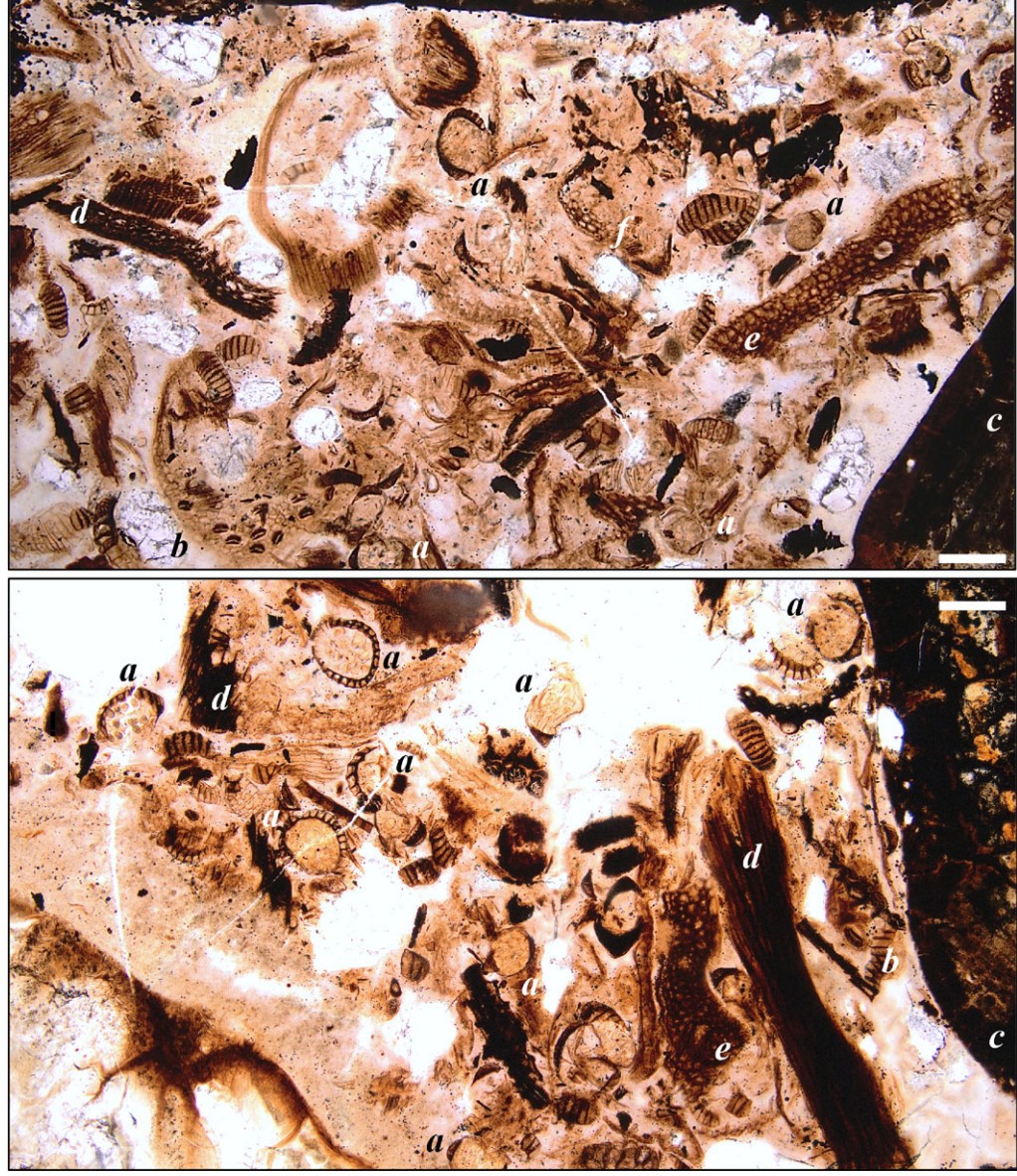

**Figure 5.** Wide views (top and bottom panels) showing abundance of plant material found in the histology slides of the cololite sample. In both, (*a*) sporangia, (*b*) leaf cuticle with stomata present, (*c*) gastroliths, (*d*) woody material, (*e*) leaf cross-sections and (*f*) sclerenchyma. Top, slide 3; bottom, slide 6. Scale bars = 200 μm.

higher diversity of palynomorphs, with 42 taxa recognized, or 84% of the combined total palynomorph diversity of 50. By contrast the cololite has a lower diversity, with only 24 palynomorphs, representing just 48% of total diversity. Of the 50 palynomorph taxa, 16 (32%) were common to both samples, 26 (52%) were found only in the external matrix and 8 (16%) were found only in the cololite (electronic supplementary material, figure S2). The external matrix was rich in moss (six taxa), fern and other pteridophyte spores (24 taxa), and gymnosperm pollen (11 taxa), whereas the cololite sample had only one moss taxon, 15 pteridophyte spore taxa, five gymnosperm pollen taxa, and two angiosperm pollen taxa, and lacked cycad pollen (electronic supplementary material, appendix 1). The pteridophyte spores in the external matrix included taxa not assignable as either leptosporangiate or eusporangiate ferns (*Impardecispora* spp.) as well as lycophytes (e.g. *Retitriletes singhii*), all of which were absent from the cololite sample. All of the fern spores in the cololite sample were from leptosporangiate groups and included Osmundaceae. No araucarian pollen was identified in either sample; however, the pollen *Classopollis classoides* in the external matrix is assigned to the extinct conifer family Cheirolepidiaceae by some palynologists.

The two samples, although sharing numerous palynomorphs, are therefore somewhat distinct, with the cololite sample being largely a subset of the more diverse external matrix. This result is consistent

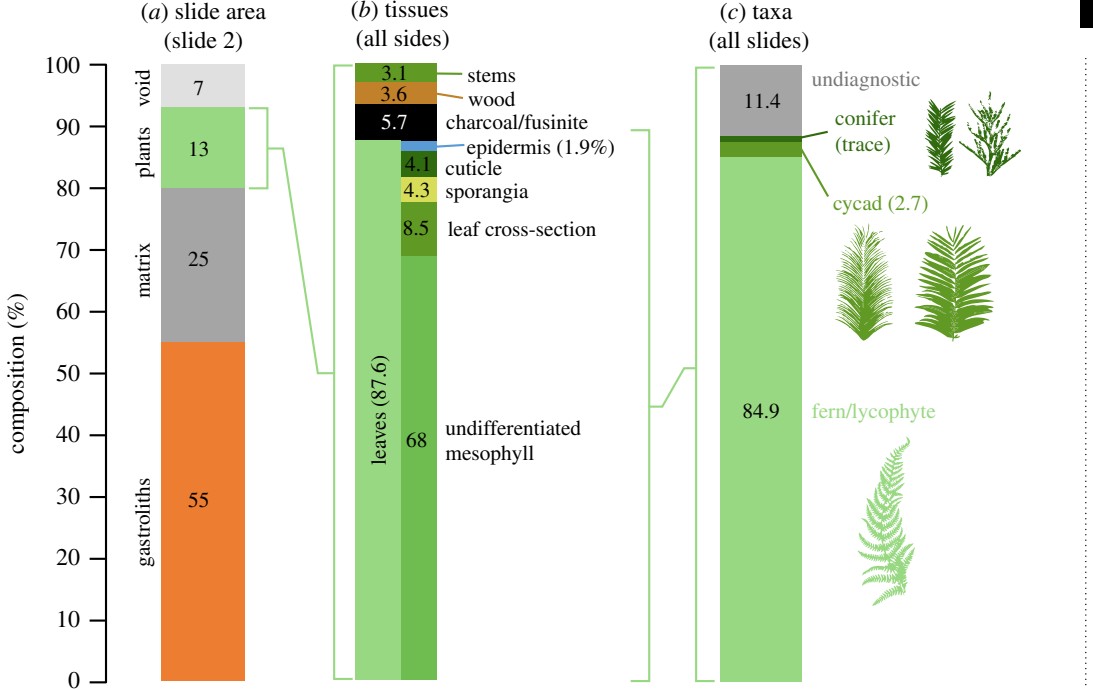

**Figure 6.** Composition of the cololite determined from microscopy of thin sections. (*a*) Breakdown of slide area (slide 2 only) occupied by gastroliths, matrix, plant fragments and void space as only slide 2 was scored for the non-plant composition. Breakdown of the plant fragment composition only (across all slides) into tissue types (*b*) and leaf specific tissue types. Breakdown of plant fragments (across all slides) into broad taxonomic groups (*c*).

with a temporally and spatially restricted sample of local flora eaten by the nodosaur versus the regional flora reflected in the external matrix, which represents sampling over an extended period of time. Consistent with the depositional environment of marine sediments, the external matrix was both rich in, and dominated by bisaccate gymnosperm pollen, including Pinaceae, Podocarpaceae and extinct taxa such as the pteridosperm *Alisporites* and *Vitreisporites pallidus* (Caytoniales).

## 3.3. Microscopic palaeobotanical analysis of stomach contents

For slide 2, where all transects and all grid squares were counted (for a total of 88 629 grid squares), 13% of grid squares contained plant fragments, whereas gastrolith material or solely matrix were present in 55% and 25% of the grid squares, respectively (figure 6*a*). A gastrolith record represents a grid square where at least half the square was gastrolith. For some fields of view, all grid squares were occupied by gastrolith material.

*Plant fragments*—Most of the material was macerated, probably through a combination of biting/shearing, and subsequent processing through the grinding action of the gastroliths in the animal's stomach (gastric mill), yielding fragments sheared at oblique angles (figures 4 and 5), obscuring cell detail. The presence of fern sporangia with *in situ* spores (figure 4*b*–*d*) is not typically seen in sediment samples and supports interpretation of the mass as a cololite. The sporangia also allowed for some narrowing of our taxonomic identification of remains (the dominant source of leaf material in the samples) to leptosporangiate ferns (subclass Polypodiidae; [76]) such as Schizaeaceae, but excluding Osmundaceae (e.g. *Osmunda*), and excluding eusporangiate ferns such as Marattiaceae (e.g. *Angiopteris*). Sporangium type C (figure 4*a*) contained *Echinatisporis* grains, indicating a clubmoss (cf. Lycopodiaceae or Selaginellaceae).

The plant fragments counted on the slides (figure 6*b,c*; electronic supplementary material, S3) showed a dominance in the nodosaurid's gut contents of leaf tissue (88%), composed of 9% recognizable leaf cross-sections (figure 4*h*) and 69% undifferentiated plant material that was interpreted as primarily leaf mesophyll owing to the predominance of parenchyma in this tissue, with some differentiation into likely densely packed palisade and spongy-mesophyll tissues (figure 4*h*). About 1% of the counted leaf fragments included prominent multi-cell-thick sclerenchyma tissue (figure 4*k*). Rare cuticle fragments (4%) preserved the pattern of the underlying leaf epidermis cells, with some of these fragments (1%) bearing stomata with distinctive flanges at the poles of the guard cells (figure 4*i,j*) that are diagnostic of

some extant cycads [77,78]. While at very low counts, some cuticle fragments bore stomata comparable to conifers, ferns or angiosperms (e.g. Figure 4*l*). Non-leaf plant fragments were composed of stem cross-sections (3%) sometimes showing annual growth rings (figure 4*m*), rare other plant structures (e.g. strobili), and woody twig cross-sections and highly fragmented wood material (4%) (figure 6).

At least five different isolated leptosporangiate fern sporangia morphotypes were recognized (table 3 and figure 4*b–d*), but were only 4.3% of the total count and were often in the same grid square as the cuticle types 1 and 2. While not taxonomically diagnostic, fern epidermis is often composed of cells with highly sinuous lateral walls; 3% from the total of 4% cuticle counted possessed this trait, with three morphotypes (types 1–3) recognized on the basis of the differing frequencies of the sinuosity (figure 4*g,i*).

Of interest was the presence (6%) of blackened plant fragments that sometimes preserve anatomical detail (figure 4*e*). These fragments we interpret as burnt plant material (charcoal, inertinite or fusinite).

Although variation exists, the relative abundance of major plant tissue types is consistent across the slides (electronic supplementary material, figure S3), suggesting that the inclusions within the cololite matrix are fairly homogeneous, and the results are reflective of the overall pattern and not highly influenced by isolated concentrations.

## 3.4. Lower Albian Gates Formation, Grand Cache Member macroflora

Representative examples of the plant taxa from the Grand Cache Member are shown in figure 3. Ferns (class Polypodiopsida) were well represented in the macroflora, with five genera present: *Acanthopteris* and *Coniopteris* (Dicksoniaceae s.l.); *Gleichenites* (Gleicheniaceae); and *Cladophlebis* (Osmundales) and *Pseudophlebis* (fam. indet.). The seed fern *Sphenopteris* was also present. Dicksoniaceae and Gleicheniaceae are leptosporangiate ferns and Osmundales are basal leptosporangiate ferns (subclass Polypodiidae), while the others are of uncertain taxonomic placement. *Equisetites* (cf. *Equisetum*) was prevalent in the Grand Cache Member [74]. Gymnosperms such as Caytoniales (e.g. *Sagenopteris*) were numerous and conifers (Cupressales–Pinales) were diverse, with the most abundant conifers *Pityophyllum* (Pinaceae), *Athrotaxites* and *Elatides* (Cupressaceae), and *Elatocladus* (incertae sedis). Cycads (Cycadales) were rare, with three genera (*Chilinia*, *Ctenis* and a newly proposed genus [74]) from only 13 specimens; however, cycadophytes (Bennettitales) were well represented in the flora, with 11 species identified by Wan [74] and numerous specimens recorded of *Ptilophyllum* s.l., *Pseudocycas* and *Pterophyllum*. *Ginkgo* and *Ginkgoites* (Ginkgoales) and *Taeniopteris* (incertae sedis) were relatively abundant. Czekanowskiales (extinct ginkgophytes) also made up a small portion of the flora, with only 1 genus and 12 specimens. Angiosperms were scarce in the Grand Cache Member with perhaps only three species represented by four unidentified angiosperm leaves [74].

# 4. Discussion

## 4.1. Attribution of abdominal mass to dietary stomach contents (cololite)

Multiple independent lines of evidence support the interpretation of the abdominal mass of the *Borealopelta markmitchelli* specimen TMP 2011.033.0001 as a cololite (ingested stomach contents), and not peri- or postdepositional sediment infill. Of the 16 criteria developed to evaluate putative stomach contents, the specimen meets 14 (table 2; electronic supplementary material, table S1).

In total, the *B. markmitchelli* abdominal mass presents more criteria supporting the cololite interpretation (14) than any other purported dinosaur stomach remains (electronic supplementary material, table S1). Relative to the well-supported cololite of the *Kunbarrasaurus ieversi* specimen QM F18101 [55,56] which meets 10 criteria, the present specimen scores for these same 10, and additionally scores for association with gastroliths, distinct margins, distinct palynomorphs internal and external to the cololite, and unusual concentrations (intact sporangia).

In all aspects, the position of the cololite in the present specimen is nearly identical to that of the *Kunbarrasaurus ieversi* specimen QM F18101 [55,56]. In both cases, the cololite is located at the thoracosacral transition, even with the anterior margin of ilium, positioned to the left of the midline, and appressed dorsally near or on the ribs (figure 1*c,f*). Independently, these two cases provide unequivocal evidence of a cololite, and the similarity of the positions of these masses provided further supporting evidence. That abdominal masses of *Kunbarrasaurus* and *Borealopelta* are preserved on the left side of the body is not a trivial detail, as asymmetry in visceral organs, with the stomach positioned on the left, is

both ancestral and highly conserved within vertebrates [79,80]. The position of the cololite in *B. markmitchelli* is also broadly similar to that seen in the *Isaberrysaura mollensis* specimen MOZ-Pv 6459, although few details of the position and extent of this mass in that specimen were reported [57].

Both extant birds and crocodilians show a developmental regionalization of the stomach into an anterior glandular stomach (proventriculus), and a posterior muscular stomach (ventriculus/gizzard), with the extant phylogenetic bracket predicting a similar regionalization in the stomach of non-avian dinosaurs [80–82]. The abundance and concentration of gastroliths associated with the cololite in *B. markmitchelli* therefore not only allow for inferences that the putative gut contents belong to the stomach but specifically to the posterior stomach, which was probably modified into a muscular gizzard.

## 4.2. Previous hypothesis of Ankylosauria diet

Historically, the diet of Ankylosauria was considered to be dominated by, or exclusively composed of, plant material [83–85], although see [60,86,87] for alternative interpretations. Due to a combination of skull morphology and relatively small and simple teeth, many authors also proposed that this diet was restricted to soft, non-abrasive plant material [88–90]. This paradigm was altered by a combination of dental microwear and biomechanical work, which suggested ankylosaurs had more complex chewing mechanics [28,91–95] and may have been able to handle tougher vegetation.

Despite the research into their palaeoecology, relatively few hypotheses about the specifics of ankylosaur diet have been put forward. Weishampel [96] suggested fleshy components of cycad inflorescences as well as cycadophyte (e.g. *Nilssonia*), seed-fern (Caytoniales: e.g. *Sagenopteris*), and (in the Late Cretaceous) angiosperm fruits. Farlow [97], suggested the foliage of conifers and ferns. Tiffney [40] suggested the tough foliage of conifers and cycadophytes, and softer foliage of seed ferns, pteridophytes, and other gymnosperms, as well as cycad and conifer seeds, and (in the Late Cretaceous) angiosperms.

In the broader context of dinosaur megaherbivory, several authors have analysed the digestibility and energy content of modern relatives of potential Mesozoic dinosaur food plants [12,17,19]. Hummel *et al.* [17] suggested that horsetails, the non-leptosporangiate ferns *Angiopteris* (Marattiaceae) and *Osmunda* (Osmundaceae), the conifer family Araucariaceae and other non-podocarpaceous conifers, and *Ginkgo*/ ginkgophytes would have been good to excellent sources of food energy for herbivorous dinosaurs, while cycads, conifers of the family Podocarpaceae (e.g. *Podocarpidites multesimus*; electronic supplementary material, appendix 1) and ferns of the family Dicksoniaceae (e.g. *Coniopteris*; figure 3) would have been the least attractive. Combining these results with palaeobotanical and co-occurrence data, Gee [18] suggested *Araucaria*, *Equisetum*, the Cheirolepidiaceae and *Ginkgo* would have been the most likely food sources for sauropod dinosaurs, while conifers such as the Podocarpaceae, Cupressaceae and Pinaceae would be less optimal, but still viable food. Ferns such as *Angiopteris* and *Osmunda*, as well as cycads and cycadophytes (Bennettitales: e.g. *Pterophyllum*; figure 3), were considered less likely for fully grown sauropods. Regardless of their poor energetic and nutritional content, Dodson [98] suggested the ubiquity of ferns means they probably formed a major component of herbivorous dinosaur diet. Although largely in the context of sauropod diet, many of these results are also probably applicable to megaherbivorous ornithischian taxa, including ankylosaurs, other than distinctions due to feeding height limitations.

Very little data are available concerning the vegetation of Alberta contemporaneous with *Borealopelta*. The Lower Albian Grand Cache Member of the Gates Formation [74], however, provides a picture of the local vegetation potentially available to *Borealopelta* and other herbivorous dinosaurs (figure 3). This macroflora is rich in leptosporangiate ferns (15 species; e.g. Osmundales, Dicksoniaceae and Gleicheniaceae), horsetails (*Equisetites*), conifers (eight species; principally Cupressaceae and Pinaceae) and other gymnosperms such as Caytoniales (six species), ginkgophytes (Ginkgoales–Czekanowskiales; three species) and cycads–cycadophytes (Cycadales–Bennettitales; 13 species). The Grand Cache Member of the Gates Formation also preserves abundant dinosaur tracks, with the ichnotaxon *Tetrapodosaurus borealis* (Ankylosauria) being most common, further strengthening the potential association with the contemporaneous *Borealopelta* [99]. Additionally, the mollusk *Murraia naiadiformis* (Unionidae), previously only known from the upper McMurray Formation [100], has also been found in the Gates Formation [101] (e.g. TMP 1997.070.0015), also suggesting a link between these two depositional environments.

When assessing the plant groups found in the Grand Cache Member of the Gates Formation, based on their desirability as food sources, this assemblage appears to be of relatively low diet quality. Gee [18] suggested that *Araucaria*, *Equisetum*, Cheirolepidiaceae and *Ginkgo* are plants of high dietary desirability. Of these only *Ginkgo/Ginkgoites* and *Equisetites* were present, while Araucariaceae and Cheirolepidiaceae were absent. The less attractive, but still desirable food sources, Cupressaceae and Pinaceae (two species each) and *Elatocladus* (fam. indet., three species) are the principal conifers in the Gates Formation

macroflora. Lastly, the food sources least preferred [18] are the non-leptosporangiate fern *Angiopteris* and the basal leptosporangiate fern *Osmunda* (although based on the presumed limited accessibility of closed forests to large sauropods), as well as cycads, and Bennettitales. Of these groups, only Cycadales and Bennettitales were present in the Gates Formation flora, although *Cladophlebis* is often placed in the Osmundales [102] together with *Osmunda* (Osmundaceae).

## 4.3. Direct evidence of Ankylosauria diet based on *Borealopelta*

Any discussion of dietary reconstruction based on this specimen requires two caveats. Firstly, the data are based on a single specimen, which may not be representative of the species, or larger taxonomic groups as a whole. Secondly, the data represent a single brief event, probably of the order of hours, and at the end of one individual animal's life, and may not accurately reflect the typical or average diet of the individual nor the taxon, especially in the context of seasonal changes and landscape variation in food availability. These caveats aside, these data do represent the best available direct evidence of diet in an herbivorous non-avian dinosaur.

The diet of the Early Cretaceous nodosaurid ankylosaur *Borealopelta markmitchelli* was dominated (approx. 88%) by leaf material, with only a minor stem/twig component (approx. 7%) (figure 6; electronic supplementary material, table S2). This is consistent with the hypothesis that nodosaurs were selective feeders [103,104], analogous to extant large mammal herbivores such as cervids, which crop leaves and ingest only a minor fraction of smaller diameter twigs [105], and indicating feeding choices at the scale of individual plants or bites. Conifer and cycad–cycadophyte leaf fragments were rare (although potentially under-recognized), with the leaf fraction dominated by ferns (greater than 80%). There was no definitive evidence of angiosperm leaf material, although some rare cuticle fragments bore stomata resembling those seen in 'dicot' angiosperms. Angiosperms were probably rare in the local landscape at this time, as only two angiosperm pollen species were recorded in the stomach content sample (electronic supplementary material, appendix 1), and only three, potentially dubious, species of angiosperm leaves are present as rare components in the Albian Gates Formation flora [74].

Stem cross-sections (3%) probably represent small twigs of conifers, some showing evidence of 2–3 years of wood annual rings (figure 4*m*). Highly fragmented woody material and tracheids (4%) are probably derived from twigs processed by the nodosaurid's gastric mill or by its teeth and rhamphotheca. Identification of the 'stem cross-section square' material (less than 1%) as leaf petioles (cycad–cycadophyte or conifer), conifer needles or fern rachides—as observed for other records of ankylosaur gut contents and putative ankylosaur coprolites [45,56]—is possible, as rare specimens have the pattern of vascular bundles consistent with this interpretation. There is no evidence of horsetails in this nodosaurid's diet, as extant *Equisetum* species have a diagnostic stem anatomy (polygonal in cross-section with vascular bundles in a ring around a central pith or hollow centre, often displaying peripheral carinal canals) that was not observed in any of the stem fragments. Nor were *Equisetum* spores recognized in any palynological preparations. The absence of *Equisetum* on the slides is contrary to some hypotheses about horsetails as well as ferns being an important component of thyreophoran diets [12]. While *Equisetites* was recognized in the Gates Formation macroflora (figure 3*i*) and this group occurs commonly in floras throughout the Cretaceous, its absence in the cololite may simply reflect a lack of *Equisetites* in the area grazed by *Borealopelta* on the days before its death. Alternatively, the digestibility of *Equisetum* is high [17,18], perhaps selectively removing *Equisetites* from the cololite. However, as *Equisetum* fragments were identifiable in the faeces of extant geese and grizzly bears that fed on *Equisetum* [106,107], it is unlikely to have been absent from the cololite if ingested.

Leaf fragments with multiseriate sclerenchyma below the epidermis (figure 4*l*) are interpreted as cycads–cycadophytes, as this sclerenchymatous tissue by location and character is typical of the hypodermis of extant cycads, for example *Cycas circinalis*. Conifer leaves can also have a sclerenchymatous hypodermis, but where present (e.g. *Araucaria angustifolia*) the hypodermis is only one or two cells thick, even at the leaf midrib [108]. Possible pollen of the extinct Mesozoic conifer family Cheirolepidiaceae (*Classopollis classoides*) was recorded from the external matrix only (electronic supplementary material, appendix 1), but the distinctive stomata of *Frenelopsis* [109] were not observed on any of the rare cuticle fragments, and Cheirolepidiaceae are absent from the Gates Formation macroflora. The occurrences on the slides of gymnosperm leaf fragments partially correspond to the pollen records from the cololite sample, with conifer (e.g. *Taxodiaceaepollenites vacuipites* = a taxodioid Cupressaceae; and the bisaccate grains *Pityosporites* spp. and *Podocarpidites multesimus* = Cupressales or Pinales) and angiosperm pollen present (e.g. *Tricolpites* sp. and *Tricolporites* sp.). Cycad pollen (i.e. *Cycadopites formosus*), however, was only present in the external matrix sample (electronic supplementary material, appendix 1).

The detection of cycad–cycadophyte leaf fragments in the cololite slides, and only a trace of conifer leaves on the slides, is contrary to previous prediction of diet choice for large herbivorous dinosaurs (although based on sauropods, animals much larger than *Borealopelta*), which argued against cycads and favoured araucarian leaves based on the energetics of their digestion [12,110]. Adult sauropods, however, would have accessed trees, whereas *Borealopelta* would have accessed low-stature plants. Conifers and *Ginkgo* are common in the Gates Formation macroflora (figure 3*d,f,g*), and these taxa form medium to tall trees, elevating the foliage high above ground level, making conifers and *Ginkgo* less likely food sources for the low-placed heads of nodosaurids, except when these gymnosperms were present as saplings. By contrast, cycad–cycadophytes include both tree-like forms and low-statured plants (including trunkless zamioids) and were also common in the Gates Formation macroflora (figure 3*a*). The very low counts of cycad–cycadophytes in the slides (3%) (including both leaf fragments with hypodermis (figure 4*k*) and cuticle fragments with stomata) is probably an underestimate, as the undifferentiated plant tissue category was associated with cuticle with cycad-like stomata (e.g. Figure 4*i,j*) in very low counts and so may represent cycad–cycadophyte leaf tissue lacking sclerenchyma. More commonly though, the undifferentiated plant tissue was associated with likely fern cuticle, often with fern sporangia present, consistent with fern leaf tissue being the strongly dominant component of the undifferentiated category. Our interpretation, therefore, is that cycad–cycadophyte leaves were a minor component in the nodosaurid's diet (less than 5%) and may represent casual ingestion. Similarly, the significant presence of plant fragments as charcoal or fusinite (6%) is strongly suggestive of consumption by *Borealopelta* of plants that had sent out new shoots after recent local wildfire.

Leptosporangiate ferns (identified in the cololite slides through the presence of sporangia with a well-developed annulus) include the fern orders Cyatheales, Gleicheniales, Polypodiales and Schizaeales (figure 4*b–d*). In addition to the leptosporangiate sporangia were isolated sporangia still containing identifiable trilete grains of a clubmoss, *Echinatisporites* (figure 4*a*). Both leptosporangiate (e.g. *Cyathidites* spp./Cyatheales, *Cicatricosporites* spp./Schizeaeales and *Laevigatosporites haardti*/Polypodiales) and Osmundaceae (Osmundales) spores (e.g. *Baculatosporites comaumensis* and *Osmundacidites wellmanii*) are recognized in the cololite palynology sample (electronic supplementary material, appendix 1), although no sporangia were observed with *B. comaumensis* or *O. wellmanii* spores *in situ*. These data are consistent with the animal not consuming Osmundaceae/*Osmunda* or non-leptosporangiate fern leaves (e.g. *Angiopteris*).

Experimental data from the fermentation of living analogues of Mesozoic flora have predicted that horsetails (*Equisetum*/*Equisetites* spp.), the non-leptosporangiate ferns *Angiopteris* (Marattiaceae) and *Osmunda* (Osmundaceae), the conifer family Araucariaceae and other non-podocarpaceous conifers, and *Ginkgo*/*Ginkgoites* would have been good to excellent sources of food energy for herbivorous dinosaurs such as thyreophorans, while cycads, conifers of the family Podocarpaceae (e.g. *Podocarpidites multesimus*) and ferns of the Dicksoniaceae (e.g. *Coniopteris*) would have been the least attractive ([12] and references therein). Contrary to this prediction, this ankylosaur individual selected leptosporangiate ferns, including *Biretisporites*-producing plants (aff. Matoniaceae–Cyatheaceae–Dicksoniaceae), Schizaeaceae (e.g. *Cicatricosisporites* sp.) and probably also Gleicheniales (e.g. *Gleicheniidites* sp.), over conifers and basal-leptosporangiate ferns such as *Osmunda*/*Cladophlebis* or non-leptosporangiate ferns such as *Angiopteris* (figure 4). Our data are equivocal on the importance of cycads–cycadophytes (e.g. *Ctenis*, *Nilssonia*, *Pterophyllum*) in its diet, but are consistent with cycads–cycadophytes being a minor component at best. As animals with their head held close to the ground (less than 1 m), a diet rich in ferns, the dominant low-to-the-ground plants (i.e. less than 50 cm), is not surprising. Based on modern analogues, most conifers (e.g. Cupressaceae, Pinaceae) and *Ginkgo* tend to have few branches below 1 m, except for saplings. If present, low-statured conifers and cycads–cycadophytes appear to have been passed over or incidentally consumed. These results suggest that flora components that are energetically less favourable may still make up a significant portion of the diet of large dinosaur herbivores, and that digestive energetics alone may not predict dietary consumption in non-analogue megaherbivores. The results further highlight the dramatic, and non-analogue, dietary and ecological differences between Mesozoic dinosaur megaherbivores and extant mammal megaherbivores.

The woody stem cross-section (figure 4*m*) shows distinct growth rings, with the outermost ring incomplete, consistent with the twig being consumed mid-growing season, probably in late spring to mid-summer, timing consistent also with the presence of clubmoss and fern sporangia with mature spores (figure 4*a–d*).

## 4.4. Forest fire succession

The presence of burnt plant fragments in the stomach contents, and its abundance (6%) relative to non-burnt wood tissue (4%), suggest a relatively high amount of charcoal or fusinite was consumed.

Although charcoal is known to be intentionally consumed by multiple wild animals (see review in [111]), it may also be incidentally consumed during feeding in an area recently subject to wildfire. In this case, it is unclear if the consumption is intentional or incidental, but regardless indicates that the environment occupied by the animal prior to death was recently subject to wildfire.

Growing evidence suggests that forest fire was a common event within Early Cretaceous conifer-dominated forests globally [112,113]. On a more regional scale, abundant evidence for forest fire in the form of fossil charcoal also exists for the Lower Albian Gates Formation and Manville Group of northern British Columbia and Alberta [112–115], which are approximately contemporaneous with, and in geographical proximity to, *Borealopelta*. Indeed, wildfire may have been a frequent and regular occurrence in the depositional environment now represented by the Gates Formation, possibly in the frequency of once every 20–40 years [115].

Modern ferns often play an important role in post-fire succession in both tropical [116–118] and temperate [119–121] forests. Due to the scarcity and low diversity of angiosperms at this time and abundance of conifers, ferns may have played an even larger role in fire succession during the Early Cretaceous. Given that the animal's diet was strongly dominated by fern foliage, and the potential for ferns to dominate distinct phases of post-fire succession, forests recovering from wildfire may have represented distinct niches ideal for herbivorous, low-browsing dinosaurs, such as ankylosaurs.

Analysis of habitat usage by extant mammalian ungulates has documented regional preferences for recently burned and recovering areas among many species, both in the context of conifer-dominated forests ([122] and references therein) and grasslands ([123] and references therein). The data from the analysed stomach contents, and the context from modern analogues, suggest that, like many modern herbivores, *Borealopelta* may also have preferred recently burned areas to take advantage of post-fire succession. There are several factors which may have resulted in these areas being desirable, including abundant low-lying plants (in this case, ferns), high concentrations of nutrients in palatable regrowth, and a more open and/or heterogeneous habitat. If this interpretation is correct, the dietary contents of the nodosaur represents the earliest evidence of a fire succession usage within a large-bodied herbivore. On a wider scale, cycles of wildfire and post-fire succession in Cretaceous gymnosperm-dominated forests may have been important palaeoecological factors for many Cretaceous animals, including other herbivorous dinosaurs.

Ethics. The study did not involve humans or live animals, and does not require associated ethics approval.

Data accessibility. The datasets supporting this article have been uploaded as part of the electronic supplementary material.

Authors' Contributions. C.M.B., D.R.G., D.M.H. and J.F.B. conceived of and designed the study, and D.R.G. and C.M.B. coordinated the study. C.L.G. collected the primary data from the palaeobotanical histological counting and took the images with contributions from J.E.K. C.M.B., D.M.H. and D.R.G. participated in data analysis, and drafted the manuscript with contributions from all of the authors. J.E.K., D.R.G. and J.F.B. collated the data and images from the Gates Formation leaf fossils with assistance from C.M.B. D.R.B. identified and analysed the palynological samples. All authors participated in interpretation of the results, gave final approval for publication, and agree to be held accountable for the work performed therein.

Competing interests. We have no competing interests.

Funding. The Palaeoecology Laboratory microscopy suite at Brandon University was funded from successive grants from the Canada Foundation for Innovation and Research Manitoba to D.R.G. and in-kind support from Olympus Canada. Partial funding for this research was through a NSERC Discovery grant to D.R.G., funding from the University of Saskatchewan to J.F.B., National Geographic Society Grant to C.M.B., and funding from the Royal Tyrrell Museum of Palaeontology, Royal Tyrrell Museum Cooperating Society to J.E.K. and Suncor Canada.

Acknowledgements. We thank Kirstin Brink, Donald Brinkman, Kentaro Chiba, David Eberth, Paul Johnston, Mark Mitchell, Dan Spivak, François Therrien and Jakob Vinther for fruitful discussions. François Therrien provided microscope access for initial examination. Sue Sabrowski, Jakob Vinther and Mike Eklund aided in specimen photography. Raymond Strom and Stephen Cheung of Calgary Rock and Materials Services prepared histological thin sections. Palynomorph samples were prepared by Russ Harms and Global Geolab Ltd. Donna Sloan produced the scientific illustration in figure 1*b*. CT scans were performed at Western Veterinary Specialist and Emergency Centre, Calgary, with the assistance of Nic Roussset, Joni Klaassen and Cathy Gaviller. Specimen access and assistance was provided by Tom Courtenay, Heather Feeney, Warren Nicholls, Rhian Russell, Becky Sanchez and Brandon Strilisky. Carole Gee and Atilla Ősi provided reviews that significantly improved the quality of the paper, and Allison Dailey, Kevin Padian and Anita Kristiansen handled the submission.

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
