## [Reviewer comments · Royal Society Open Science]

Review History

RSOS-200305.R0 (Original submission)

Review form: Reviewer 1 (Atilla Osi)

Is the manuscript scientifically sound in its present form?

Yes

Are the interpretations and conclusions justified by the results?

Yes

Is the language acceptable?

Yes

Do you have any ethical concerns with this paper?

No

Have you any concerns about statistical analyses in this paper?

No

Recommendation?

Accept with minor revision (please list in comments)

Comments to the Author(s)

This manuscript is about the stomach content of an exceptionally preserved armored dinosaur, unique to the history of dinosaur research. There is no doubt about that it is a stomach content; 14 of the 16 criteria supporting this are fulfilled for this finding. And the content of the stomach itself preserved the remains of the last meal in such detail that we can have a very detailed picture of what this herbivorous animal have consumed. After many decades of research, there is finally a herbivorous dinosaur finding where much of the food consumed can be identified, in some cases, at the genus level. The discovery contradicts some earlier theories that we can learn a lot about the feeding mode and food preference of ornithischian dinosaurs.

The manuscript is well written, well legible and covers every detail. The most important available methods were used to examine the material in question.

The illustrations are informative and show all relevant details.

Besides a few small comments, I fully support the publication of this manuscript.

Comments:

A small but not negligible part of the stomach contents contains charred plant residues. The authors interpret this as saying that this herbivorous dinosaur may have been grazing various ferns in an area where fire had previously destroyed the forest. This allowed the charred pieces of plant to enter the animal's stomach.

However, it is well known that in many vertebrate animals, including domestic animals, sometimes consume carbonized plant material, which contributes to their better digestion and, among others, the binding of harmful substances. For pets, Schmidt et al. (2019, PeerJ) compiled all relevant work on the topic, which is an excellent demonstration of the need for vegetable charcoal from time to time for many animals, not just herbivores. (In the case of domestic animals, this is added as an additive to certain foods).

I suggest the authors to discuss the possibility that this animal may have consciously consumed these pieces of carbonized plant remains, perhaps in the context of its digestion. Since we knew the last food that this animal consumed, could this not be related to the fact that this animal was possibly ill, may have had e.g. digestive problems and therefore had taken the charred pieces of plant?

- Please, check figure citations in the text. For example, on p. 5. in the last paragraph figure 3 h-j is cited two times in relation to glauconite component and a few lines later to the stomach wall, but figure 3 is about the Grand Cache Member flora.

- In the first paragraph of the Discussion, table 1 is cited but I guess authors wanted to cite table 2.

- In chapter 5.2. some references are cited in relation to complex chewing mechanics. I suggest to add or change ref. 92 to Ósi et al. (2016, Hist. Biol.), since this work gives more update and diverse picture on ankylosaur jaw mechanism and feeding.

- P. 11 line 15. nodoosaurid's diet. ->nodosaurid's diet

- On figure 5, the explanation for label „D” is missing.

Budapest, 16.03.2020.

Sincerely,
Attila Ósi

Review form: Reviewer 2 (Carole Gee)

Is the manuscript scientifically sound in its present form?

Yes

Are the interpretations and conclusions justified by the results?

No

Is the language acceptable?

Yes

Do you have any ethical concerns with this paper?

No

Have you any concerns about statistical analyses in this paper?

No

Recommendation?

Accept with minor revision (please list in comments)

Comments to the Author(s)

I wish to reveal my identity as a reviewer, because I think it is important to discuss science openly. Anyway, I have only good things to say about this paper, with the exception of the misinterpretation and misapplication of the results from our work (Hummel et al. 2008; Gee 2011), which likely stem from a too-quick reading of our papers or a lack of deeper understanding of how we came to our rating of the various plant groups as possible sauropod food plants. However, these issues can be cleared up with a bit of rewriting. I discuss these points and offer suggestions for improvement in the attached pdf (Appendix A). I look forward to continuing the good research into as well as the interesting conversation on food plants in herbivorous dinosaurs.

- Carole Gee

Decision letter (RSOS-200305.R0)

28-Apr-2020

Dear Dr Brown

On behalf of the Editors, I am pleased to inform you that your Manuscript RSOS-200305 entitled "Dietary palaeoecology of an Early Cretaceous armoured dinosaur (Ornithischia; Nodosauridae) based on floral analysis of stomach contents" has been accepted for publication in Royal Society Open Science subject to minor revision in accordance with the referee suggestions. Please find the referees' comments at the end of this email.

The reviewers and handling editors have recommended publication, but also suggest some minor revisions to your manuscript. Therefore, I invite you to respond to the comments and revise your manuscript.

- Ethics statement

- Data accessibility

<http://datadryad.org/submit?journalID=RSOS&manu=RSOS-200305>

- Competing interests

- Authors' contributions

- Acknowledgements

- Funding statement

Because the schedule for publication is very tight, it is a condition of publication that you submit the revised version of your manuscript before 07-May-2020. Please note that the revision deadline will expire at 00.00am on this date. If you do not think you will be able to meet this date please let me know immediately.

To revise your manuscript, log into <https://mc.manuscriptcentral.com/rsos> and enter your Author Centre, where you will find your manuscript title listed under "Manuscripts with

Decisions". Under "Actions," click on "Create a Revision." You will be unable to make your revisions on the originally submitted version of the manuscript. Instead, revise your manuscript and upload a new version through your Author Centre.

If your manuscript is newly submitted and subsequently accepted for publication, you will be asked to pay the article processing charge, unless you request a waiver and this is approved by Royal Society Publishing. You can find out more about the charges at <https://royalsocietypublishing.org/rsos/charges>. Should you have any queries, please contact openscience@royalsociety.org.

on behalf of Professor Allison Daley (Associate Editor) and Kevin Padian (Subject Editor)
openscience@royalsociety.org

Associate Editor Comments to Author (Professor Allison Daley):
Comments to the Author:

The reviewers have positively assessed the research of this manuscript, agreeing with the authors that this does represent a real stomach content, and commending the high quality methods and results assessing the diet of *Borealopelta*. The authors are requested to address the minor changes suggested by the reviewers to the text of the discussion, specifically by considering the purposes of ingestion of charcoal, and by modifying the direct comparison of the diet of *Borealopelta* to the diet of huge sauropods by incorporating more consideration of the size and stature difference (and vegetation they could easily reach) of each. Otherwise, please thoroughly check all call-outs to Figures, Tables and references, and include the minor changes to the text suggested by both reviewers.

Reviewer comments to Author:
Reviewer: 1

Comments to the Author(s)

This manuscript is about the stomach content of an exceptionally preserved armored dinosaur, unique to the history of dinosaur research. There is no doubt about that it is a stomach content; 14 of the 16 criteria supporting this are fulfilled for this finding. And the content of the stomach itself preserved the remains of the last meal in such detail that we can have a very detailed picture of what this herbivorous animal have consumed. After many decades of research, there is finally a herbivorous dinosaur finding where much of the food consumed can be identified, in some cases, at the genus level. The discovery contradicts some earlier theories that we can learn a lot about the feeding mode and food preference of ornithischian dinosaurs.

The manuscript is well written, well legible and covers every detail. The most important available methods were used to examine the material in question.

The illustrations are informative and show all relevant details.

Besides a few small comments, I fully support the publication of this manuscript.

Comments:

A small but not negligible part of the stomach contents contains charred plant residues. The authors interpret this as saying that this herbivorous dinosaur may have been grazing various ferns in an area where fire had previously destroyed the forest. This allowed the charred pieces of plant to enter the animal's stomach.

However, it is well known that in many vertebrate animals, including domestic animals, sometimes consume carbonized plant material, which contributes to their better digestion and, among others, the binding of harmful substances. For pets, Schmidt et al. (2019, PeerJ) compiled all relevant work on the topic, which is an excellent demonstration of the need for vegetable

charcoal from time to time for many animals, not just herbivores. (In the case of domestic animals, this is added as an additive to certain foods).

I suggest the authors to discuss the possibility that this animal may have consciously consumed these pieces of carbonized plant remains, perhaps in the context of its digestion. Since we knew the last food that this animal consumed, could this not be related to the fact that this animal was possibly ill, may have had e.g. digestive problems and therefore had taken the charred pieces of plant?

- Please, check figure citations in the text. For example, on p. 5. in the last paragraph figure 3 h-j is cited two times in relation to glauconite component and a few lines later to the stomach wall, but figure 3 is about the Grand Cache Member flora.

- In the first paragraph of the Discussion, table 1 is cited but I guess authors wanted to cite table 2.

- In chapter 5.2. some references are cited in relation to complex chewing mechanics. I suggest to add or change ref. 92 to Ósi et al. (2016, Hist. Biol.), since this work gives more update and diverse picture on ankylosaur jaw mechanism and feeding.

- P. 11 line 15. nodosaurid's diet. ->nodosaurid's diet

- On figure 5, the explanation for label „D” is missing.

Budapest, 16.03.2020.

Sincerely,
Attila Ósi

Reviewer: 2

Comments to the Author(s)

I wish to reveal my identity as a reviewer, because I think it is important to discuss science openly. Anyway, I have only good things to say about this paper, with the exception of the misinterpretation and misapplication of the results from our work (Hummel et al. 2008; Gee 2011), which likely stem from a too-quick reading of our papers or a lack of deeper understanding of how we came to our rating of the various plant groups as possible sauropod food plants. However, these issues can be cleared up with a bit of rewriting. I discuss these points and offer suggestions for improvement in the attached pdf

(Review_of_Brown_et_al_Royal_Open_Science.pdf). I look forward to continuing the good research into as well as the interesting conversation on food plants in herbivorous dinosaurs.

- Carole Gee

Author's Response to Decision Letter for (RSOS-200305.R0)

See Appendix B.

Decision letter (RSOS-200305.R1)

18-May-2020

Dear Dr Brown,

It is a pleasure to accept your manuscript entitled "Dietary palaeoecology of an Early Cretaceous armoured dinosaur (Ornithischia; Nodosauridae) based on floral analysis of stomach contents" in

its current form for publication in Royal Society Open Science. The comments of the reviewer(s) who reviewed your manuscript are included at the foot of this letter.

on behalf of Professor Allison Daley (Associate Editor) and Kevin Padian (Subject Editor)
openscience@royalsociety.org

Associate Editor Comments to Author (Professor Allison Daley):

Comments to the Author:

Thank you to the authors for thoroughly and thoughtfully revising the manuscript based on the reviewers suggestions. I am happy to recommend accepting this paper as is.

Appendix A

Review of Brown et al., Dietary palaeoecology of an Early Cretaceous armoured dinosaur (Ornithischia; Nodosauridae) based on floral analysis of stomach contents

Manuscript ID: RSOS-200305

For *Royal Society Open Science*, 2020

This manuscript is a well-written and careful study of ancient digestive remains—the cololite, or stomach contents—found in a well-preserved Early Cretaceous armored dinosaur. The fossil material is spectacular, the approach of the study is good, and the identification of plant remains from the cololite is solid. Together with the uniqueness of this fossil occurrence, this study represents a meaningful contribution to the literature. Thus, this well-documented study of a single meal of an armored *Borealopelta* dinosaur—its last supper, so to speak—is novel and worthy of being published.

Thank you for giving me the opportunity to review this interesting manuscript. I have been working on the dietary preferences of sauropods for over a decade now and am pleased that Brown et al. have found empirical evidence for food plants in an herbivorous dinosaur. However, there are several differences between their small-stature *Borealopelta* ankylosaur and the sauropods that we have studied. The major difference concerns size, and the ecological feeding strategies and digestive physiology associated with it. *Borealopelta* was a small-stature herbivorous dinosaur, with short legs and a short neck, and thus, its feeding height, dietary preferences, and even digestive strategies cannot be compared to those of sauropods. Furthermore, because of the relatively small body mass of *Borealopelta*, it is doubtful that it had as urgent a need

to optimize its intake of highly digestible plant matter as the much larger sauropods did. In fact, even without knowing the contents of the cololite described in Brown et al.'s study, I would have predicted *Borealopelta* fed on ferns and other low-growing plants based on its low height, downward-oriented head, short neck, and relatively small body mass. And if the feeding habitat of *Borealopelta*, which is interpreted by Brown et al. in this manuscript, was indeed one that was affected by wildfire, I would have not expected the ingestion of *Equisetum*, for this is not an environment in which horsetails are commonly found. For these reasons, I generally concur with the interpretations put forth in this manuscript.

What I disagree with is the misinterpretation and misapplication on the part of Brown et al. of our work (Hummel et al. 2008; Gee 2011). I am of the belief that their results do not conflict with ours, as they state repeatedly on pages 10 to 12, because one cannot compare the feeding height, foliage attainability, or feeding strategy of ankylosaurs to that of sauropods. Please note that all the interpretations and hypotheses made by Hummel et al. (2008) and Gee (2011) are for fully grown, adult sauropods, which must have been extreme bulk feeders, while the smaller size of *Borealopelta* would have allowed it to graze on plants that may have been less than optimally nutritious. Thus, the results of Brown et al. for this ankylosaur do not conflict with ours, nor can they be directly compared to ours for the giant sauropods.

Another major difference is that we chose to look at possible dietary preferences based on the digestibility and energy-release of the nearest living relatives of the Jurassic flora. Thus, in our approach, we necessarily missed out on the taxa that have gone extinct; a diversity of these extinct ferns is surely represented in the fossil remains in the cololite of Brown et al., for there are undoubtedly many Mesozoic plants that did not survive to the present day. Another possibility for *Borealopelta* having a fern-rich

diet is that these particular Cretaceous ferns may have also been more digestible and nutritious than those living ferns in our lab analyses. The most compelling explanation in my mind, though, for the fern-dominated diet of *Borealopelta* is that ferns were the most abundant leafy plants growing in the foraging area that this small-stature dinosaur could reach.

For these reasons, I feel that some changes in the Discussion section of the Brown et al. manuscript should be made for clarification and accuracy. I have mentioned these changes in the list below.

The following points outline my suggestions for improvement.

- Two plant names are misspelled throughout the manuscript. It should be Marattiaceae, not Marratiaceae. Similarly, it should be Cheirolepidiaceae, not Cheirolepidaceae.
- In the second paragraph of the Results section, the spelling of “diametre” and “diametres” would be more consistent than “diameter” and “diameters” given the type of English used in the rest of this manuscript (see “centimetre” in the same paragraph).
- In the second paragraph of section 4.3, shouldn’t be call-out for figures 4 and 5 have an s at the end of “figure”?
- p. 8, line 6, it would be more elegantly phrased this way: The plant fragments counted. . .
- p. 8, line 11, please check the call-out for figure 4*b*; shouldn’t it be 4*h*?
- p. 8, line 16, I don’t understand the call-out here to figure 3*b*.
- p. 8, line 18, I would add “woody” so that it reads: . . .were composed of woody stem cross sections. . .

- p. 8, line 30, again, I don't understand the call-out here to figure 3.
- p. 8, line 41: *Cladophlebis* is considered to represent the Mesozoic foliage of the fern family Osmundaceae. This is supported by the association of the fertile fossil leaves of *Todites* that are morphologically the same as those of the living *Todea* of the Osmundaceae, with *Cladophlebis* leaves, known from the Jurassic onwards.
- p. 10, line 11, add "for sauropod dinosaurs" so that it reads: . . .would have been the most likely food sources for the sauropod dinosaurs.
- p. 10, lines 15 to 16, add "for fully grown sauropods" so that it reads: . . .were considered less likely for fully grown sauropods. [Note that the reasoning offered by me in 2011 is based on the present-day preference of *Angiopteris* and *Osmunda* for closed forests, which would have been difficult habitats for full-sized adult sauropods to maneuver in; the smaller ankylosaurs would not have had this problem. In fact, if Cretaceous *Cladophlebis* was physiologically and nutritiously similar to living *Osmunda*, I'm sure it would have been one of the food plants most favored by many smaller herbivorous dinosaurs.]
- p. 10, lines 19 to 20: I completely disagree with the statement that is "these results [regarding dietary preferences of sauropods] are likely applicable to Ornithischian taxa, including ankylosaurs, . . ." As I mention earlier in this review, I think that the feeding strategies of ankylosaurs may have been completely different from those of the giant sauropods; because of their much smaller sizes, the ankylosaurs probably did not need to bulk feed on optimally nutritious foliage like the extremely large sauropods did; *Borealopelta* may have done just fine on ferns, even if these plants offered less nutrition and fewer calories per mouthful. Moreover, it is clear that *Borealopelta* could not reach the foliage on the taller conifer trees and was probably simply limited to lower growing leaves. Alternatively, ankylosaurs may have sought

out fern-dominated patches in closed forests to avoid space competition with fully grown sauropods. Nutrition, feeding strategies, and the occupation of various feeding niches were likely as complex in the Mesozoic as they are today. At any rate, I suggest deleting the entire last sentence of this paragraph because of its inaccuracy.

- p. 10, line 26, add the Osmundaceae to this list of fern families, for *Cladophlebis* is a member of this family.
- P. 10, line 42 to 46: The last two sentences in this paragraph are incorrect. The first of the two sentences is wrong due to a lack of understanding of the results and interpretations in Gee (2011). As explained above, the low likelihood of sauropods targeting *Angiopteris* and *Osmunda* was based on the habitat preference of these ferns for closed forests today, and the unlikelihood that giant sauropods could maneuver and readily forage in such a habitat; ankylosaurs would not have had this size-related problem. The second sentence excludes the Osmundaceae, which is represented in the local flora as *Cladophlebis* pinnae.
- p. 10, line 57, add “non-avian” so that it reads: . . . best available evidence of diet in a nonavian herbivorous dinosaur. [Note that the fossil seeds found in the abdomen of *Jeholornis prima*, a Cretaceous bird, are widely accepted as gut contents; see Sander et al. 2010.]
- p. 11, line 30: There are two additional reasons that may explain why *Equisetum* remains were not found in the cololite, although this genus occurs in the nearby megaf flora. One reason is that horsetails may have not been locally abundant at the *Borealopelta* feeding grounds, which are interpreted here as a fire-swept environment; fossil and living *Equisetum* prefers damp or even shallow freshwater habitats. The second is that all living species of *Equisetum* that we have tested in follow-up studies since Hummel et al. (2008) and Gee (2011) are extremely digestible

(Gee et al., manuscript in progress), and any horsetails consumed by the *Borealopelta* may have been digested to a point past recognizability in this cololite

- p. 11, line 45: Doesn't *Alisporites* typically represent seed ferns? [I don't know the species *A. bilateralis*.]
- p. 11, line 51: I beg to differ on this point once more. I do not consider *Borealopelta* to be a large animal, and I don't think that Brown et al. can compare the dietary predictions made for huge adult sauropods with what they found in a small-stature ankylosaur. Please rephrase.
- p. 11, line 54: replace "are typically tree-like" with "form medium to tall trees"
- p. 12, line 4, please check if figure 4f is the correct photo
- p. 12, lines 27 to 55. Much of the first part of this paragraph repeats what has been said about the predictions of sauropod dietary preferences in the lowermost paragraph of p. 11. I realize that this because the paragraph on p. 11 focuses on fossil leaf fragments, while that on p. 12 refers to the palynoflora. Still, it would be better to edit the writing in the first paragraph to reduce so much repetition in the manuscript. Of the two, I find that the wording and approach taken in paragraph on p. 12 are excellent, because this second paragraph very elegantly remarks on the differences between sauropod and ankylosaur feeding strategies linked to height limitations, as well as touches on the complexity of discerning dietary preferences in extinct animals.

Once again, thanks for asking me to review this very interesting study. I'm sure that after the numerous small changes that I have suggested have been made, this will be an amazing paper.

Respectfully submitted,

Carole Gee

University of Bonn, Germany

Appendix B

Dear editors,

Thank for you for the efficient handing of our recent submission to *Royal Society Open Science* entitled “Dietary palaeoecology of an Early Cretaceous armoured dinosaur (Ornithischia; Nodosauridae) based on floral analysis of stomach contents” (Manuscript ID: RSOS-200305), and for the opportunity to submit a revised version. We have gone through the reviewers’ comments and have revised the manuscript to address the concerns of Drs Gee and Ósi. We have itemized our response to each reviewer comment below. We feel that these revisions, based on the reviewers’ suggestions, have significantly improved the quality of the manuscript, and have added this recognition to our Acknowledgements.

In addition to the changes suggested by Drs Gee and Ósi, we have also made a small number of changes to the manuscript to correct minor typographical errors, clarify some language and added one previously overlooked reference. These changes are minor and do not alter our interpretation or reporting of the results. All changes to the manuscript can be seen in the Track Changes version of both the main document as well as the ESM. One additional change was an alteration of author order. This was done at the request of Donald Henderson who felt the contributions of two co-authors (Jessica Kalyniuk and Dennis Braman) exceeded his own, and should be listed before his name.

All of the authors have contributed to the revisions or at least reviewed them, and have approved the revisions and this response to reviewers.

We have exported figures from the original format to EPS (and PDF), however the field sizes for several of these figures range from 200-400 MB, well above the maximum file size allowable for upload on the SchoalrOne Manuscripts website (100 MB total). Would you be able to advise us on how to process for these figures? For the moment have resubmitted JPG version of these particular figures.

Please let us know if you have any questions or concerns,

Thank you for your time,

Caleb Brown and David Greenwood,
and on behalf of the other authors

Reviewer 1: Attila Ósi

Comments to the Author(s)

This manuscript is about the stomach content of an exceptionally preserved armored dinosaur, unique to the history of dinosaur research. There is no doubt about that it is a stomach content; 14 of the 16 criteria supporting this are fulfilled for this finding. And the content of the stomach itself preserved the remains of the last meal in such detail that we can have a very detailed picture of what this herbivorous animal have consumed. After many decades of research, there is finally a herbivorous dinosaur finding where much of the food consumed can be identified, in

some cases, at the genus level. The discovery contradicts some earlier theories that we can learn a lot about the feeding mode and food preference of ornithischian dinosaurs.

The manuscript is well written, well legible and covers every detail. The most important available methods were used to examine the material in question.

The illustrations are informative and show all relevant details.

Besides a few small comments, I fully support the publication of this manuscript.

Comments:

A small but not negligible part of the stomach contents contains charred plant residues. The authors interpret this as saying that this herbivorous dinosaur may have been grazing various ferns in an area where fire had previously destroyed the forest. This allowed the charred pieces of plant to enter the animal's stomach.

However, it is well known that in many vertebrate animals, including domestic animals, sometimes consume carbonized plant material, which contributes to their better digestion and, among others, the binding of harmful substances. For pets, Schmidt et al. (2019, PeerJ) compiled all relevant work on the topic, which is an excellent demonstration of the need for vegetable charcoal from time to time for many animals, not just herbivores. (In the case of domestic animals, this is added as an additive to certain foods).

I suggest the authors to discuss the possibility that this animal may have consciously consumed these pieces of carbonized plant remains, perhaps in the context of its digestion. Since we knew the last food that this animal consumed, could this not be related to the fact that this animal was possibly ill, may have had e.g. digestive problems and therefore had taken the charred pieces of plant?

AUTHORS: This is a valid point. This discussion in this section has been modified to reflect that we do not know whether the consumption was intentional or incidental, and this reference has been added. Regardless, much of the remaining discussion is unchanged, as the animal would still need to have encountered a recently burnt area.

The text now reads: “Although charcoal is known to be intentionally consumed by multiple wild animals [see review in 106], it may also be incidentally consumed during feeding in an area recently subject to wildfire. In this case, it is unclear if the consumption was intentional or incidental, but regardless indicates that the environment occupied by the animal prior to death was recently subject to wildfire.”

- Please, check figure citations in the text. For example, on p. 5. in the last paragraph figure 3 h-j is cited two times in relation to glauconite component and a few lines later to the stomach wall, but figure 3 is about the Grand Cache Member flora.

AUTHORS: Yes, this was incorrect and should refer to figure S3 (rather than figure 3). It has now been corrected throughout the document.

- In the first paragraph of the Discussion, table 1 is cited but I guess authors wanted to cite table 2.

AUTHORS: Yes, this was incorrect. It has now been corrected from “table 1&S1” to “table 2&S1”.

- In chapter 5.2. some references are cited in relation to complex chewing mechanics. I suggest to add or change ref. 92 to Ósi et al. (2016, *Hist. Biol.*), since this work gives more update and diverse picture on ankylosaur jaw mechanism and feeding.

AUTHORS: This reference has now been added.

- P. 11 line 15. *nodoosaurid's diet.* ->*nodosaurid's diet*

AUTHORS: This correction has been made.

- On figure 5, the explanation for label „D” is missing.

AUTHORS: The explanation for labels B_C were misplaced in the caption, and D omitted. This error has been corrected.

We thank Dr Ósi for his thoughtful and careful review of our submission. Several useful corrections were incorporated in the revised version.

Reviewer 2: Carole Gee

This manuscript is a well-written and careful study of ancient digestive remains—the cololite, or stomach contents—found in a well-preserved Early Cretaceous armored dinosaur. The fossil material is spectacular, the approach of the study is good, and the identification of plant remains from the cololite is solid. Together with the uniqueness of this fossil occurrence, this study represents a meaningful contribution to the literature. Thus, this well-documented study of a single meal of an armored Borealopelta dinosaur—its last supper, so to speak—is novel and worthy of being published.

Thank you for giving me the opportunity to review this interesting manuscript. I have been working on the dietary preferences of sauropods for over a decade now and am pleased that Brown et al. have found empirical evidence for food plants in an herbivorous dinosaur. However, there are several differences between their small- stature Borealopelta ankylosaur and the sauropods that we have studied. The major difference concerns size, and the ecological feeding strategies and digestive physiology associated with it. Borealopelta was a small-stature herbivorous dinosaur, with short legs and a short neck, and thus, its feeding height, dietary preferences, and even digestive strategies cannot be compared to those of sauropods. Furthermore, because of the relatively small body mass of Borealopelta, it is doubtful that it had as urgent a need to optimize its intake of highly digestible plant matter as the much larger sauropods did. In fact, even without knowing the contents of the cololite described in Brown et al.'s study, I would have predicted Borealopelta fed on ferns and other low-growing plants based on its low height, downward-oriented head, short neck, and relatively small body mass. And if the feeding habitat of Borealopelta, which is interpreted by Brown et al. in this manuscript, was indeed one that was affected by wildfire, I would have not expected the ingestion of Equisetum, for this is not an environment in which horsetails are commonly found. For these reasons, I generally concur with the interpretations put forth in this manuscript.

What I disagree with is the misinterpretation and misapplication on the part of Brown et al. of our work (Hummel et al. 2008; Gee 2011). I am of the belief that their results do not conflict with ours, as they state repeatedly on pages 10 to 12, because one cannot compare the feeding height, foliage attainability, or feeding strategy of ankylosaurs to that of sauropods. Please note that all the interpretations and hypotheses made by Hummel et al. (2008) and Gee (2011) are for fully grown, adult sauropods, which must have been extreme bulk feeders, while the smaller size of Borealopelta would have allowed it to graze on plants that may have been less than optimally nutritious. Thus, the results of Brown et al. for this ankylosaur do not conflict with ours, nor can they be directly compared to ours for the giant sauropods.

Another major difference is that we chose to look at possible dietary preferences based on the digestibility and energy-release of the nearest living relatives of the Jurassic flora. Thus, in our approach, we necessarily missed out on the taxa that have gone extinct; a diversity of these extinct ferns is surely represented in the fossil remains in the cololite of Brown et al., for there are undoubtedly many Mesozoic plants that did not survive to the present day. Another possibility for Borealopelta having a fern-rich diet is that these particular Cretaceous ferns may have also been more digestible and nutritious than those living ferns in our lab analyses. The most compelling explanation in my mind, though, for the fern-dominated diet of Borealopelta is that ferns were the most abundant leafy plants growing in the foraging area that this small-stature dinosaur could reach.

For these reasons, I feel that some changes in the Discussion section of the Brown et al. manuscript should be made for clarification and accuracy. I have mentioned these changes in the list below.

The following points outline my suggestions for improvement.

- Two plant names are misspelled throughout the manuscript. It should be Marattiaceae, not Marratiaceae. Similarly, it should be Cheirolepidiaceae, not Cheirolepidaceae.

AUTHORS: This has now been corrected.

- In the second paragraph of the Results section, the spelling of “diametre” and “diametres” would be more consistent than “diameter” and “diameters” given the type of English used in the rest of this manuscript (see “centimetre” in the same paragraph).

AUTHORS: No change. According to the Oxford and Cambridge University dictionaries diameter/diameters is the correct spelling in UK English, Canadian English, and Australian English, all of which use the spelling centimetre and centre, vs. centimeter and center in US English.

- In the second paragraph of section 4.3, shouldn't be call-out for figures 4 and 5 have an s at the end of “figure”?

AUTHORS: This has now been corrected.

- p. 8, line 6, it would be more elegantly phrased this way: The plant fragments counted. . .

AUTHORS: This has now been corrected.

- p. 8, line 11, please check the call-out for figure 4b; shouldn't it be 4h?

AUTHORS: This has now been corrected.

- p. 8, line 16, I don't understand the call-out here to figure 3b.

AUTHORS: 3b was included here in error. This has now been removed.

- p. 8, line 18, I would add "woody" so that it reads: . . .were composed of woody stem cross sections. . .

AUTHORS: We have changed the wording so it better corresponds to our classification scheme (Tables 3, S2 and Fig. 6) where wood tissue (a grouped category that included woody stems as well as fragmented wood tissue) was 3.6%.

- p. 8, line 30, again, I don't understand the call-out here to figure 3.

AUTHORS: This was an error, and reference to fig. 3 has been removed.

- p. 8, line 41: *Cladophlebis* is considered to represent the Mesozoic foliage of the fern family Osmundaceae. This is supported by the association of the fertile fossil leaves of *Todites* that are morphologically the same as those of the living *Todea* of the Osmundaceae, with *Cladophlebis* leaves, known from the Jurassic onwards.

AUTHORS: In recent accounts of Mesozoic ferns we have consulted, *Cladophlebis* is attributed to the Osmundales, but not to the family Osmundaceae owing to the lack of sporangial characters diagnostic of the family. We have made this change and have added citation of a recent source for *Cladophlebis* being Osmundales.

- p. 10, line 11, add "for sauropod dinosaurs" so that it reads: . . .would have been the most likely food sources for the sauropod dinosaurs.

AUTHORS: This edit has been made as suggested.

- p. 10, lines 15 to 16, add "for fully grown sauropods" so that it reads: . . .were considered less likely for fully grown sauropods. [Note that the reasoning offered by me in 2011 is based on the present-day preference of *Angiopteris* and *Osmunda* for closed forests, which would have been difficult habitats for full-sized adult sauropods to maneuver in; the smaller ankylosaurs would not have had this problem. In fact, if Cretaceous *Cladophlebis* was physiologically and nutritiously similar to living *Osmunda*, I'm sure it would have been one of the food plants most favored by many smaller herbivorous dinosaurs.]

AUTHORS: This edit has been made as suggested.

- p. 10, lines 19 to 20: I completely disagree with the statement that is "these results [regarding

dietary preferences of sauropods] are likely applicable to Ornithischian taxa, including ankylosaurs, . . .” As I mention earlier in this review, I think that the feeding strategies of ankylosaurs may have been completely different from those of the giant sauropods; because of their much smaller sizes, the ankylosaurs probably did not need to bulk feed on optimally nutritious foliage like the extremely large sauropods did; Borealopelta may have done just fine on ferns, even if these plants offered less nutrition and fewer calories per mouthful. Moreover, it is clear that Borealopelta could not reach the foliage on the taller conifer trees and was probably simply limited to lower growing leaves. Alternatively, ankylosaurs may have sought out fern-dominated patches in closed forests to avoid space competition with fully grown sauropods. Nutrition, feeding strategies, and the occupation of various feeding niches were likely as complex in the Mesozoic as they are today. At any rate, I suggest deleting the entire last sentence of this paragraph because of its inaccuracy.

AUTHORS: We have re-worded the sentences as so: “Although largely in the context of sauropod diet, many these results are also applicable to megaherivorous Ornithischian taxa, including ankylosaurs, other than distinctions due to feeding height limitations.”

- p. 10, line 26, add the Osmundaceae to this list of fern families, for *Cladophlebis* is a member of this family.

AUTHORS: consistent with our statement above, we have added Osmundales to this list.

- P. 10, line 42 to 46: *The last two sentences in this paragraph are incorrect. The first of the two sentences is wrong due to a lack of understanding of the results and interpretations in Gee (2011). As explained above, the low likelihood of sauropods targeting *Angiopteris* and *Osmunda* was based on the habitat preference of these ferns for closed forests today, and the unlikelihood that giant sauropods could maneuver and readily forage in such a habitat; ankylosaurs would not have had this size-related problem. The second sentence excludes the Osmundaceae, which is represented in the local flora as *Cladophlebis pinnae*.*

AUTHORS: The caveat ‘(although based on hypothesized limited accessibility to closed forest by large sauropods)’ has been added to this section to clarify the first point. The second sentence has also been adjusted to read “Of these groups, only Cycadales and Bennettitales were present in the Gates Formation flora, although *Cladophlebis* is often placed in the Osmundales [102] together with *Osmunda* (Osmundaceae).” to take into account the reviewers other comment.

- p. 10, line 57, add “non-avian” so that it reads: . . . best available evidence of diet in a nonavian herbivorous dinosaur. [Note that the fossil seeds found in the abdomen of *Jeholornis prima*, a Cretaceous bird, are widely accepted as gut contents; see Sander et al. 2010.]

AUTHORS: This edit has been made as suggested

- p. 11, line 30: *There are two additional reasons that may explain why *Equisetum* remains were not found in the cololite, although this genus occurs in the nearby megafloora. One reason is that horsetails may have not been locally abundant at the Borealopelta feeding grounds, which are*

interpreted here as a fire-swept environment; fossil and living Equisetum prefers damp or even shallow freshwater habitats. The second is that all living species of Equisetum that we have tested in follow-up studies since Hummel et al. (2008) and Gee (2011) are extremely digestible (Gee et al., manuscript in progress), and any horsetails consumed by the Borealopelta may have been digested to a point past recognizability in this cololite

AUTHORS: We do comment in the manuscript that rarity of *Equisetites* in the feeding grounds at the time of *Borealopelta*'s last meal may have contributed to its absence in the cololite. Pg 11 lines 28-31. However, we have amended this statement to more clearly express this concept.

Regarding a lack of preservation within the cololite explaining the absence of *Equisetites*, we find this an ad hoc and implausible suggestion. On a quick review of literature, we found some studies of extant birds and mammals that routinely fed on *Equisetum*, and while these studies concur with the finding for high digestibility of *Equisetum*, these studies found that *Equisetum* fragments were readily identifiable in these animals' feces. Geese, like our nodosaur, have gastroliths. If extant geese, grizzly bear, and caribou scat preserve *Equisetum* stems with their more advanced chemical and microbial digestion vs. a nodosaurid, Occam's razor leads to an expectation that had *Borealopelta* eaten *Equisetum* it would be recognizable in the cololite. We have added a sentence to make this point.

- p. 11, line 45: *Doesn't Alisporites typically represent seed ferns? [I don't know the species A. bilateralis.]*

AUTHORS: We have removed *Alisporites* from the text at this point and corrected it to be listed as a pteridosperm in Appendix 1.

- p. 11, line 51: *I beg to differ on this point once more. I do not consider Borealopelta to be a large animal, and I don't think that Brown et al. can compare the dietary predictions made for huge adult sauropods with what they found in a small-stature ankylosaur. Please rephrase.*

AUTHORS: We disagree that it is incorrect to refer to *Borealopelta* as a "large herbivore". This is because conservative body mass estimates place *Borealopelta* at 1,300 kg, well above the 1,000 kg threshold used to define the megaherbivore guild (NB: defined in the 1st sentence of the 2nd paragraph of the Introduction), and *Borealopelta* is the largest documented herbivorous taxon within both its host formation and the geographically proximate penecontemporaneous formations. We do concede that *Borealopelta* is smaller than many of the sauropod dinosaurs that were the subject of the Gee 2011 paper. To address Dr Gee's concerns, we have clarified that while *Borealopelta* as a large herbivore, it is smaller than those for which these inferences were made. The sentence now reads, with an added follow-up sentence:

"The detection of cycad–cycadophyte leaf fragments in the cololite slides, and only a trace of conifer leaves on the slides, is contrary to previous prediction of diet choice for large herbivorous dinosaurs (although based on sauropods, much larger animals than *Borealopelta*), which argued against cycads and favoured araucarian leaves based on the energetics of their digestion [12, 110]. Adult sauropods, however, would have accessed trees, whereas

Borealopelta would have accessed low-stature plants.”

- p. 11, line 54: replace “are typically tree-like” with “form medium to tall trees”

AUTHORS: Change made.

- p. 12, line 4, please check if figure 4f is the correct photo

AUTHORS: Change made to cite 4i,j.

- p. 12, lines 27 to 55. Much of the first part of this paragraph repeats what has been said about the predictions of sauropod dietary preferences in the lowermost paragraph of p. 11. I realize that this because the paragraph on p. 11 focuses on fossil leaf fragments, while that on p. 12 refers to the palynoflora. Still, it would be better to edit the writing in the first paragraph to reduce so much repetition in the manuscript. Of the two, I find that the wording and approach taken in paragraph on p. 12 are excellent, because this second paragraph very elegantly remarks on the differences between sauropod and ankylosaur feeding strategies linked to height limitations, as well as touches on the complexity of discerning dietary preferences in extinct animals. Once again, thanks for asking me to review this very interesting study. I’m sure that after the numerous small changes that I have suggested have been made, this will be an amazing paper.

AUTHORS: As we have made changes to the text on p. 11 as above to address the concerns raised by Dr Gee, we believe the change proposed here is moot. The detail presented at lines 27 to 55 on p. 12 we feel sets up the discussion that follows and is necessary for the flow of our argument.

We thank Dr Gee for her careful and thoughtful review. Some useful improvements and corrections have resulted, and we look forward to Dr Gee reading the final published version and appreciate her expectation that this will be ‘an amazing paper’.

Caleb Brown and David Greenwood,
and on behalf of the other authors